# Natural climate variability is an important aspect of future projections of snow water resources and rain-on-snow events

Michael Schirmer[1,2], Adam Winstral[2†], Tobias Jonas[2], Paolo Burlando[3], Nadav Peleg[3,4]

[1]Swiss Federal Institute for Forest, Snow and Landscape Research, 8903 Birmensdorf, Switzerland
[2]WSL Institute for Snow and Avalanche Research SLF, 7260 Davos, Switzerland
† Deceased in March, 2021
[3]Institute of Environmental Engineering, ETH Zurich, 8093 Zurich, Switzerland
[4]Institute of Earth Surface Dynamics, University of Lausanne, 1015 Lausanne, Switzerland

*Correspondence to*: Michael Schirmer (michael.schirmer@wsl.ch)

**Abstract.**

Climate projection studies of future changes in snow conditions and resulting rain-on-snow (ROS) flood events are subject to large uncertainties. Typically, emission scenario uncertainties and climate model uncertainties are included. This is the first study on this topic to also include quantification of natural climate variability, which is the dominant uncertainty for precipitation at local scales with large implications for e.g. runoff projections. To quantify natural climate variability, a weather generator was applied to simulate inherently consistent climate variables for multiple realizations of current and future climates at 100 m spatial and hourly temporal resolution over a 12 x 12 km high-altitude study area in the Swiss Alps. The output of the weather generator was used as input for subsequent simulations with an energy balance snow model. The climate change signal for snow water resources stands out as early as mid-century from the noise originating from the three sources of uncertainty investigated, namely uncertainty in emission scenarios, uncertainty in climate models, and natural climate variability. For ROS events, a climate change signal toward more frequent and intense events was found for an RCP 8.5 scenario at high elevations at the end of the century, consistently with other studies. However, for ROS events with a substantial contribution of snowmelt to runoff (>20%), the climate change signal was largely masked by sources of uncertainty. Only those ROS events where snowmelt does not play an important role during the event will occur considerably more frequently in the future, while ROS events with substantial snowmelt contribution will mainly occur earlier in the year but not more frequently. There are two reasons for this: first, although it will rain more frequently in midwinter, the snowpack will typically still be too cold and dry and thus cannot contribute significantly to runoff; second, the very rapid decline in snowpack toward early summer, when conditions typically prevail for substantial contributions from snowmelt, will result in a large decrease in ROS events at that time of the year. Finally, natural climate variability is the primary source of uncertainty in projections of ROS metrics until the end of the century, contributing more than 70% of the total uncertainty. These results imply that both the inclusion of natural climate variability and the use of a snow model, which includes a physically-based processes representation of water retention, are important for ROS projections at the local scale.

# 1    Introduction

The future decrease of snow depth and snow water equivalent in mountainous environments due to global warming has been
shown in several studies (e.g. Musselman et al., 2017; Marty et al., 2017; Verfaillie et al., 2018, Willibald et al., 2020). The
frequency and intensity of rain-on-snow (ROS) events are also foreseen to alter due to changes in the snow cover, the
precipitation phase, and the rain frequency and intensity (e.g. Beniston and Stoffel, 2016). Despite a decreasing snow cover,
ROS events have been predicted to become more frequent and intense at high elevations (Surfleet and Tullos, 2013; Beniston
and Stoffel, 2016; Morán-Tejeda et al., 2016; Musselman et al., 2018; Ohba and Kawase, 2020; Sezen et al., 2020). A contrary
study found that ROS events as a cause of annual runoff maxima will disappear at lower elevations and slightly decrease at
higher elevations by the end of the century (Chegwidden et al., 2020). They analyzed only annual runoff maxima, identifying
this as a key difference in methodology from Musselman et al. (2018) which may cause the difference in findings. Furthermore,
process-based hydrological models were used to investigate ROS events, thus encompassing a wider range of processes than
the former studies, which were limited to the coincidence of snow and rain. When analyzing historic observations, Sikoska-
Senoner and Seibert (2020) found a decreasing number of ROS events also in high elevated catchments.

Different sources of uncertainty were considered in some of these ROS studies, however, the relative importance of internal
climate variability compared to other uncertainty sources has not been previously assessed. The latter is largely a consequence
of the chaotic nature of the atmosphere (Deser et al., 2012a). It is a result of purely periodic external forcing, a non-linear
interplay of feedbacks within the climate system, and random fluctuations in physical or chemical factors in the atmosphere
(Ghil, 2002). For climate change analyses, the role of internal climate variability on projections of air temperature and
precipitation has been quantified together with other uncertainty sources, e.g. emission scenario and climate model uncertainty
(Hawkins and Sutton, 2009; 2011; Deser et al., 2012b, Fatichi et al., 2016, Lehner et al., 2020). In general, the smaller the
scale and the shorter the time horizon of the projections, the more important is the relative contribution of internal climate
variability to overall uncertainty (e.g. Hawkins and Sutton, 2011). Projections of precipitation are generally more affected by
natural climate variability than those of air temperature (Hawkins and Sutton, 2009; 2011; Peleg et al., 2019). For mean and
extreme precipitation at local scales (i.e. weather stations) internal climate variability is the dominant source of uncertainty,
not only for short time horizons but also through the end of this century (Fatichi et al., 2016). While it is possible for future
research to reduce the amount of uncertainty if climate models are improved or emission scenarios are constrained, the amount
of natural climate variability is not reducible. These findings raise the question of how informative climate projections based
only on climate model outputs are and will be at local scales (Fatichi et al., 2016).

Willibald et al. (2020) studied the effects of internal climate variability on the change of mean and maximum snow depth at
eight stations in the Swiss Alps and concluded that it is a major source of uncertainty for time horizons up to 50 years and
more. The effects of internal climate variability on projected runoff have been highlighted in several studies. For instance, the
climate change signals for the mean, frequency, and seasonality of runoff in the middle of this century are masked by natural
climate variability (Fatichi et al., 2014), while they will emerge by the end of the century (Addor et al., 2014). The signal

varies with elevation and is dependent on the hydrological components (e.g. snowmelt, evapotranspiration) that drive runoff (Moraga et al., 2021). Lafaysse et al. (2014) concluded that internal climate variability is capable of exacerbating, moderating, or even reversing a climate change signal of streamflow. These studies indicate the importance of including internal climate variability in studies of climate change impacts on catchment-scale hydrologic response.

In this study, the uncertainty of future projection of snow water resources and rain-on-snow characteristics at local scales were quantified in relation to natural climate variability, climate model and scenario uncertainty at the local scale. We hypothesize that snow water resources are less affected by internal climate variability than rainfall-driven runoff because they are more dependent on air temperature. The frequency and intensity of ROS events are hypothesized to be more influenced by natural precipitation variations compared to snow water equivalent (SWE), as they may be less dependent on air temperature. The

research questions are as follows:

- How important is internal variability for future projections of snow resources and rain-on-snow events?
- When is the time of emergence of changes in snow resources and rain-on-snow events?

We will explore whether the commonly found increase in ROS frequency and intensity hold for future climates when natural climate variability is considered. To this end we used simulations of a high-resolution weather generator, AWE-GEN-2D,

generating multiple stochastic ensembles of future climate projections that have been shown to realistically represent natural climate variability (Peleg et al., 2017). To account for the complexity of snow accumulation and melt processes and their response to a changing climate (in line with the discussion in Clark et al., 2016) we used in our analysis an energy balance snow model at high spatial and temporal resolution.

## 2    Methods

### 2.1    Study area

The "Gletsch" area in central Switzerland, with altitudes between 1400 and 3500 m a.s.l. and with an extent of 144 km$^2$ (Fig. 1), has been selected as the study area. It has a mean annual air temperature of -1.3°C and a mean annual precipitation of 1700 mm. Nival conditions prevail at these elevations today, yet the area is low enough that climate change may affect the current snow regime (e.g. Marty et al., 2016). The study area was chosen to encompass the elevation range for which an increase in the

90 number of ROS events has been shown in other studies. Observational data for training the weather generator and validating the model chain in and near the study area were available as detailed in Table 1. Note that the Rhone glacier is located within the study area, but its receding effect was not considered. In this study, we do not intend to investigate the combined effects of snow and glacier retreat on mean snow water resources or ROS properties, but only the climatic effects on seasonal snow.

The study area can therefore be considered more as an example area, as opposed to modelling the situation in-situ.

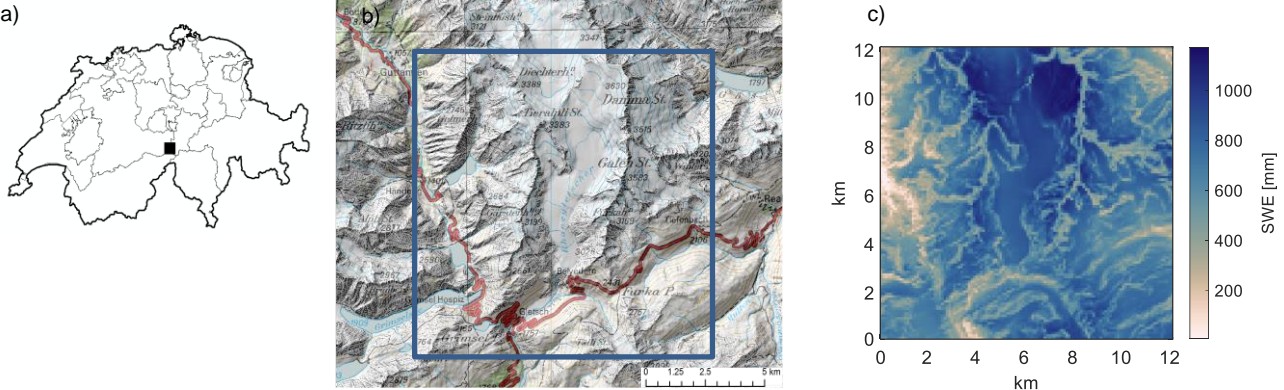

**Figure 1. Location of the study area in Switzerland (46.56° N, 8.36° E, WGS 84) (a) and map showing the extent of the model domain (source: Federal Office of Topography swisstopo) (b). Example of modeled SWE at April 1 of a random year during current climate conditions (c).**

## 2.2    Modelling setup

The model chain consists of a two-dimensional weather generator and an energy balance snow model (squares in Fig. 2). The data used or delivered by these models (ellipses in Fig. 2 are described in the following subsections).

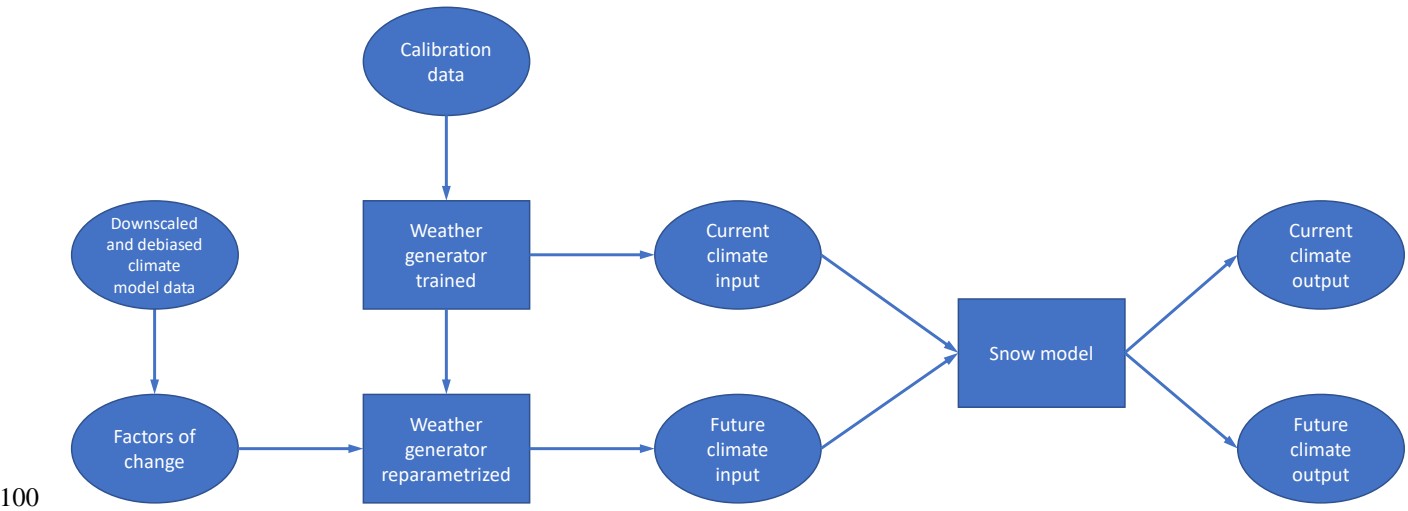

**Figure 2. Flow chart of the modelling setup.**

### 2.2.1 Climate model data

Regional climate models from the EURO-CORDEX archive (Jacob et al., 2014) were used to obtain the CH2018 climate scenarios (CH2018 Project Team, 2018), which was used in this study to calculate factors of change (FC) (Anandhi et al., 2011) needed to re-parameterize the weather generator AWE-GEN-2d in order to generate downscaled ensembles of future climate variables (cp. Sect. 2.2.2). The 10 EURO-CORDEX model chains with the highest spatial resolution of 11 km were used (Table S1). Factors of change were calculated following Peleg et al. (2019) for mean temperature, mean and variance of precipitation intensity based on seasonal projections (3 months average of the climate models), and for precipitation occurrence based on annual projections (see Appendix). The FC consists of gridded values for precipitation (cf. Figure 3b in Peleg et al., 2019) and a single value for the entire model region for temperature. They were calculated for two emission scenarios (i.e. RCP 4.5 and RCP 8.5) and two time horizons (i.e. a mid-century period from 2030 to 2059 and an end-of-century period from 2070 to 2099). A control period of 30 years (1981-2010) was used to compute the FC. Finally, FC were linearly interpolated to our 100 m resolution.

### 2.2.2 Weather generator

The AWE-GEN-2D model (Peleg et al., 2017) was used to stochastically generate gridded climate variables for the study area at 100 m spatial and hourly temporal resolution. The model was developed to simulate climate variables in complex terrain by combining physical and stochastic-statistical methods that enable preserving physical and observed dependencies between climate variables. The weather generator is capable of reproducing both principal climate statistics and the natural climate variability for the climate variables needed for subsequent energy balance snow modeling. A short description of the model structure is brought here, the readers are referred to the paper by Peleg et al. (2017) where the model and its equations are described in detail. The model first simulates a time series of dry and wet periods based on a simple renewal process, then simulates the cloud cover and precipitation (together) for each wet time step, and the cloud cover during dry periods based on the time passes from/to the closest wet period. Wind speed and direction are then simulated independently and enable the 2-dimensional advection of the precipitation fields. The near-surface air temperature is next simulated, conditioned on the cloud cover of each time step. Shortwave radiation is also directly dependent on cloud cover and on the relative humidity and dew-point temperature, which are simulated as an iterative procedure with the near-surface air temperature and vapor pressure at each time step. The longwave radiation is last computed for each time step, based on the cloud cover and near-surface air temperature.

The weather generator requires observational data for calibration, which are summarized in Table 1. Different sets of parameters are assign for each month to consider the seasonality. The spatial structure of precipitation fields, the areal intensity and the wet fraction of precipitation are calibrated using the radar data at fine space-time scales. The storm renewal process is calibrated based on precipitation data from the Grimsel station, also at fine temporal scale. Correction to the precipitation intensities, to reduce errors due to high uncertainties in the radar estimation, are conducted at the grid cell scale using

MeteoSwiss RhiresD product. In general the calibration procedure follows the procedure presented in Peleg et al. (2017); two important adjustments were made to ensure a realistic input for the energy balance modelling: first, the filter used in AWE-GEN-2d to account for orographic precipitation effects was adjusted to overcome the typical problem of undercatch by rain gauges in mountainous terrain. For this purpose, the methods described by Magnusson et al. (2014) were used to assimilate daily snow depth sensor data into the Swiss gridded precipitation product RhiresD (Scharb, 2000; MeteoSwiss, 2019). With optimal interpolation, a precipitation partitioning method, and a daily gridded temperature field (see Magnusson et al. 2014 for details), the solid precipitation fraction was adjusted. The final product are fields of total precipitation in a 1 km resolution for more than 20 years for whole Switzerland. This final product also benefits from the much denser station network of snow depth sensors at high elevations in Switzerland compared to the rain gauge network used for RhiresD. The weather generator used these gridded fields to model the spatial distribution of total precipitation on an annual basis. Second, the wind speed was spatially adjusted to match the debiased wind speeds of a numerical weather prediction model in this region (Winstral et al., 2017).

The weather generator is used in two ways, first in the trained setup with the above-mentioned data as input to generate current climate conditions, and second in a reparametrized setup using a factors of change (FC) (see section 2.2.1) approach to generate future climate conditions (Peleg et al., 2019). FCs directly affects air temperature, precipitation occurence and intensity Moreover, when these climate variables are reparameterized, they indirectly influence other variables based on the interdependencies between the variables implemented in the model (Peleg et al., 2017). Note that for generating current climate conditions, no information of the regional climate models was used (Fig. 2; Peleg et al., 2017). For both setups, a spatial resolution of 100 m was chosen to account for small-scale processes that are imperative for capturing the spatial variability of snowmelt dynamics in small mountain catchments (e.g. terrain shading of direct radiation). A resolution of 1 km was chosen for precipitation but simulations were subsequently linearly resampled to 100 m. The model domain consists of 120 x 120 grid points. 50 realizations of 30 years each, representing the same climatic period, were used to explore the natural climate variability. This is consistent with the setup presented by Peleg et al. (2017; 2019). While for future climate conditions, the weather generator was re-parameterized using factors of change (see Sect. 2.2; Peleg et al., 2019).

In summary, the weather generator was used to (1) provide hourly data for the full set of required inputs for the energy balance snow model (see next section), (2) generate climate variables with intervariable consistency, (3) downscale and debias regional climate model output, and (4) generate multiple realizations of current and future climate periods.

### 2.2.3 Snow model

The snow model used in this study is an energy balance snow model, an evolution of the Jules Investigation Model (JIM; Essery, 2013). Only a single model configuration from this multi-model framework was used, determined by comparison against comprehensive datasets including snow lysimeter data (Magnusson et al., 2015). This model was advanced by integrating a seasonal algorithm for the fraction of snow-covered area (Helbig et al., 2015; 2021), a local adjustment of the

albedo routine that better reflects the observed elevation dependency of the albedo decrease rate in Switzerland and a subgrid precipitation adjustment that takes into account the influence of topography on the distribution and redistribution of snow in mountainous terrain. Correction functions depending on aspect and slope were trained with a set of high-resolution snow depth maps from airborne LiDAR images in the European Alps as described in Grünewald and Lehning (2015). This method provides an accurate derivation of mean snow depths from snow and precipitation measurements at flat sites. This model setup is used for the Operational Snow Hydrological Service in Switzerland to predict snowmelt runoff and has been thoroughly developed through several studies (Griessinger et al., 2019; Winstral et al., 2019; Helbig et al., 2021). The snow model requires total precipitation (Precip), air temperature (TA), incoming shortwave radiation (ISWR), incoming longwave radiation (ILWR), wind, air pressure and relative humidity in an hourly resolution which was provided by the weather generator (see Sect. 2.2.2). Precipitation was split into solid and liquid phases using an adaptation of the method presented in Magnusson et al. (2014) originally developed for daily data. The snow model was run on the same resolution as the weather generator (i.e. 120 x 120 grid points with a 100 m spatial resolution).

## 2.3    Verification

The weather generator was evaluated similarly as in Peleg et al. (2017) with emphasis on precipitation extremes as this is considered as relevant to study ROS events. An example for the precipitation validation between observed data (RhiresD, i.e. single time series of 30 years, see Table 1) and simulated data (ensemble, multiple time series representing the same 30-year period), with an emphasis on the extreme precipitation intensities, for a random grid cell in the domain is illustrated in Fig. S2. Additionally, monthly values at stations within the study area (see Table 1) of air temperature (OBW 1, OBW 2), incoming longwave and shortwave radiation (GRH) were compared to the output of the weather generator.

For evaluating the ability of the energy balance model to simulate snow depth (HS) and SWE with measured input, the station GUE was selected providing all required meteorological input data for energy balance snow modeling without major gaps and in good quality during two subsequent years (see section 2.2.3). This station is located at 2286 m a.s.l. at about 13 km from the study area (see Table 1). Except for precipitation, all input data were used without any preprocessing. For precipitation, a method similar to that used to train the weather generator using optimal interpolation was chosen (Magnusson et al., 2014, see Sect. 2.2.2). Since optimal interpolation is not able to handle structural biases (i.e. site-specific undercatch in the background field), a correction factor of 1.3 (c.f. Egli et al., 2009) was chosen to correct for local undercatch and achieve better HS comparison during accumulation phases. Note that this correction factor was only used for the above point-scale simulations at GUE.

To demonstrate that the combined model chain is capable of providing reasonable HS and SWE values observed HS data and derived SWE from the OBW 2 station are available. Derived SWE was determined using observed HS and a parametric model (HS2SWE) that accumulates, compacts, and melts snow layer by layer (Magnusson et al., 2014).

For all three verification steps grid points were selected to compare them with observed station data, either by exact location when the station is located within the study area (OBW1, OBW 2, GRH), or by selecting a similar grid point (elevation, slope,

shading), if the station is outside the study area (GUE). Root mean square errors (RMSE) and an additive bias was calculated for all comparisons. The stations for validation were selected to be as close as possible to the study area and to provide all relevant data in good quality.

**Table 1. Overview of observational data used for calibration and validation. The italic inputs are weather stations either within or with shown distance to the study area. TA is air temperature, RH is relative humidity, ISWR is incoming shortwave radiation, ILWR is incoming longwave radiation, P is air pressure, Precip is total precipitation, and HS is snow depth.**

| Input | Variable | Spatial resolution | Temporal resolution | Calibration purpose | Distance in km |
|---|---|---|---|---|---|
| **Calibration** | | | | | |
| *Grimsel Hospiz (1980 m)* | Precip | Point | 10 min | (Inter-)Storm duration | 0 |
| *Engelberg (1036 m)* | TA | Point | 1 h | TA lapse rate | 18 |
| *Titlis (3040 m)* | TA | Point | 1 h | TA lapse rate | 13 |
| *Grimsel Hospiz (1980 m)* | TA, RH, ISWR | Point | 1 h | TA lapse rate, vapour pressure | 0 |
| Weather radar | Precip | 2 km x 2 km | 5 min | mean areal precipitation, wet area ratio | 0 |
| Merra-2 reanalysis | | 0.5° x 0.66° | 1 h | cloud area ratio | 0 |
| RhiresD | Precip | 2 km x 2 km | daily | mean areal precipitation | 0 |
| Optimal interpolated precipitation | Precip | 1 km x 1 km | annual | mean areal precipitation | 0 |
| | | | | | |
| **Verification** | | | | | |
| *Grimsel Hospiz (GRH, 1980 m)* | ISWR, ILWR | Point | monthly | | 0 |
| *Oberwald 1 (OBW1, 2733 m)* | TA | Point | monthly | | 0 |
| *Oberwald 2 (OBW2, 2432 m)* | TA, HS | Point | monthly | | 0 |
| *Guetsch (GUE, 2286 m)* | TA, RH, P, ISWR, ILWR, | Point | 1 h | | 13 |
| *Guetsch (GUE, 2286 m)* | HS | Point | daily | | 13 |
| RhiresD | Precip | 2 km x 2 km | daily | | 0 |

## 2.4 Rain-on-snow definition

Based on the high-resolution results, a "contributing area" of a ROS event can be defined. This procedure realistically describes the elevation-dependent effects on the phase of the precipitation in combination with the presence and condition of the snowpack. For a single ROS event, these parameters vary in space, i.e. they delineate an area of varying size that contributes significantly to a ROS event ("contributing area"). Four pixel-based criteria were applied for daily values to define a contributing area, which can be found in Table 2. The criteria differ in the amount of daily rainfall, and whether there is a substantial contribution of snowmelt to surface water input (SWI) or not. SWI is calculated with the energy balance snow model (Sect. 2.4) and is the water input available at the ground surface through either snowpack runoff, rain in case of snow-free conditions, or a mixture of both in case of fractional snowcover. Snowmelt is defined here as SWI minus rainfall, i.e. the portion of surface water input that comes from the melting process. Note that criterion 1 in Table 2 is the same as that of

Musselman et al. (2018). A "ROS day" can then be defined as a day with a contributing area exceeding a size threshold, which may depend on the application or the user. As ROS frequency we define a yearly exceedance probability as a function of the event size (see Sect. 3.2.3). In contrast, the analysis of the intensity of a ROS event and its contribution of snowmelt was only done for a predefined minimum size of a ROS event, which we chose to be 1/3 of the total area.

**Table 2. Four alternative pixel-based criteria for ROS events.**

| Criterion | SWE [mm] | Rain [mm/d] | Snowmelt [%SWI] |
|-----------|----------|-------------|-----------------|
| 1         | >10      | >10         | > 20            |
| 2         | >10      | >10         | -               |
| 3         | >10      | >20         | > 20            |
| 4         | >10      | >20         | -               |

## 2.5 Quantification of climate change in relation to uncertainty sources

For the sake of consistency, we compared only simulated values of current and future climates without analyzing climate-
225 related changes between the model and observed data. However, the model was verified against observed data under current climate conditions in Sect. 2.3. Climate period mean values of 50 and 500 (i.e. 50 realizations times ten climate models) of simulated current and future climate periods, respectively, were analyzed. The 5-95th percentile range of the 50 (500) climate period mean values was chosen to quantify natural climate variability (and climate model uncertainty for future climate conditions, respectively), consistent with other studies (e.g. Fatichi et al., 2016; Peleg et al., 2019). Note that this procedure
does not quantify the natural interannual variability (e.g. a high snow year vs. a low snow year), but how different entire climate periods are (e.g. a high snow climate period vs. a low snow climate period).

## 2.6 Uncertainty partitioning

To obtain the relative contribution of the investigated sources of uncertainty (i.e. natural climate variability $I$, climate model uncertainty $M$, and emission scenario uncertainty $S$) to the total uncertainty $T$, the partitioning method presented by Yip et al.
(2011) was applied (see Appendix).

## 3 Results and Discussion

## 3.1 Verification

### 3.1.1 Weather generator

Peleg et al. (2017) showed for a nearby mountainous region that the weather generator can reproduce principle statistics of
240 climate variables. A similar verification to Peleg et al. (2017) was conducted. Annual precipitation achieved a comparable

quality as the calibration that was done to a nearby Alpine catchment (Peleg et al. 2017), as is expected since annual mean values are used for calibration (not shown). A comparison of daily precipitation intensities with a focus on extremes are shown in Fig. S2, which indicate that extremes are better captured than other intensities, which is considered important for an ROS study. Figure 3 shows a comparison for air temperature (TA), incoming shortwave radiation (ISWR), and incoming longwave radiation (ILWR) with measured data at two stations in the study area. Note that for Grimsel Hospiz (GRH), ISWR was indirectly used for calibration of the weather generator. Specifically, the data were used to calibrate the vapor pressure, but not for the variable itself. It should also be noted that the data availability for the stations spans only a few years and may not represent the long-term distribution well. Apart from these limitations, it can be seen that TA is slightly colder at the lower station OBW2 in AWE-GEN-2D (bias of -0.8 °C), while it is quite well represented at the higher station OBW1. ISWR is underestimated in winter months, while ILWR is overestimated in spring. However, it is our understanding that the quality of the output is sufficient to analyze deviations of simulated future climate conditions from current ones, i.e. no climate-related changes are compared between the model and observed data. Note that the range plotted is inter-year variability, in contrast to Sect. 3.2 and following, where inter-climate period variability is discussed.

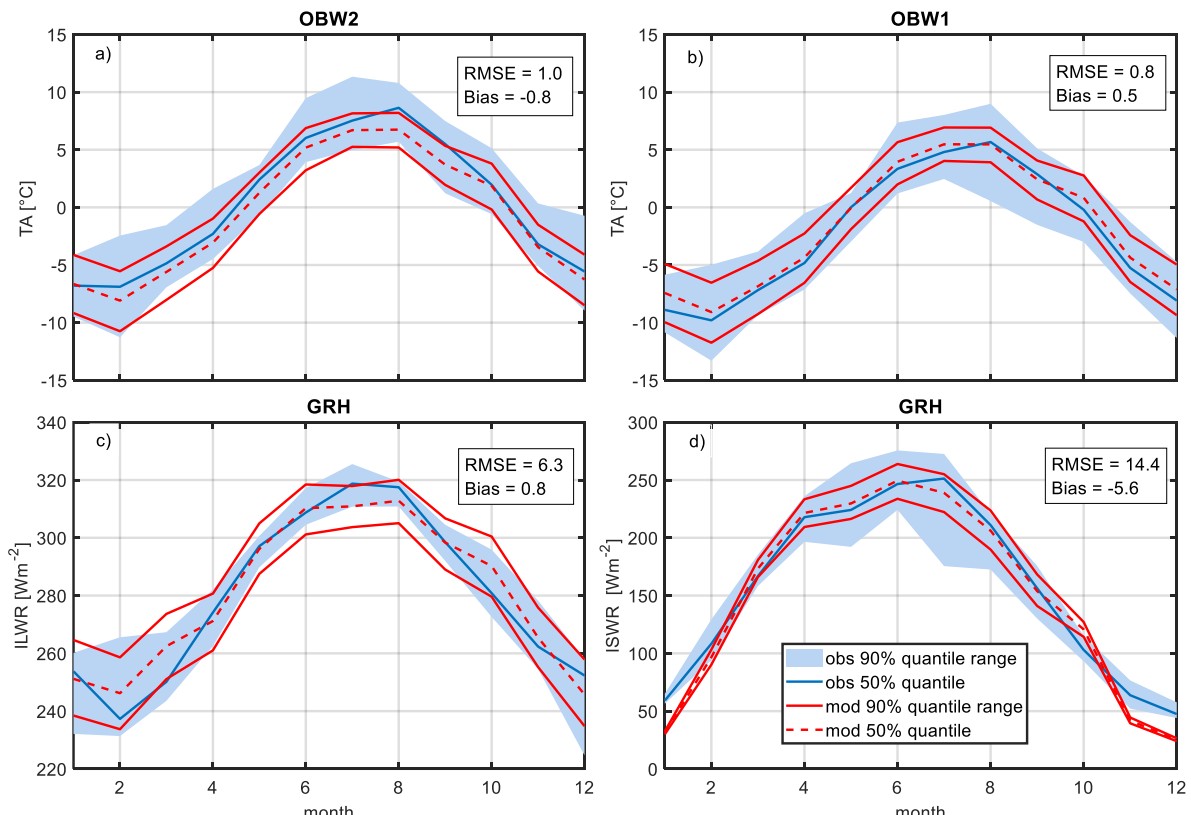

**Figure 3. Monthly mean values for (a) air temperature (TA) at the stations OBW2 and (b) OBW1, (c) incoming longwave radiation (ILWR) and (d) incoming shortwave radiation (ILWR) (d) at station GRH. Plotted are 5, 50 and 95th percentiles of observations in blue of 8 years (GRH) and 20 years (OBW1, 2), and modelled data in red of 1500 years. RMSE and Bias are calculated for 50th percentiles.**

### 3.1.2    Snow model

Recent publications demonstrate the quality of point-based snow depth modeling (Winstral et al., 2019), of spatial modelling results as inputs to a hydrologic runoff model (Griessinger et al., 2019), or in comparison to LiDAR-derived snow depth (HS) data and satellite-derived snowpack fraction data (Helbig et al., 2021). Using only measured station data as meteorological forcing, Magnusson et al. (2015) have already quantified the quality of the original JIM models with lysimeter data. In addition to these results, it is shown here that the improved model can accurately reproduce snow depth at the GUE station near the study area (Fig. 4). A good agreement was achieved in the two years studied, with an RMSE of 20 cm and a positive bias of 13 cm, calculated for days when either the model or the observations show positive snow depth (Fig. 4).

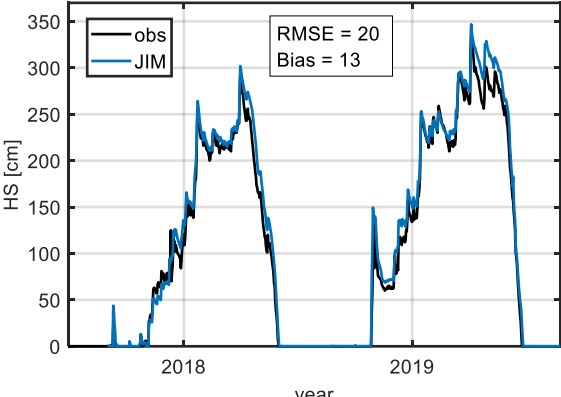

**Figure 4. HS observed (obs) and modeled (JIM) with station input at station GUE.**

### 3.1.3    Combined verification

Figure 5 shows mean values and spread of 1500 years simulated by the model chain and (pseudo) observations of HS and SWE of 20 years. The good agreement indicates that the model chain is capable of reproducing both the interannual variability and mean properties. The comparison shows, however, a slight underrepresentation of years with early intense snowfall. Note that the range in the case of the observations is determined by minimum and maximum, compared to the 5-95th percentiles of the generated data. In addition, the model typically simulates an earlier onset of melting and subsequent slower melting is typically modeled, which compensates and finally results in a mean melt out that is consistent with observations. These small inconsistencies notwithstanding, the results show a level of performance that does not compromise the use of the model combination to study the effects of climate change based on simulated current and future climate periods.

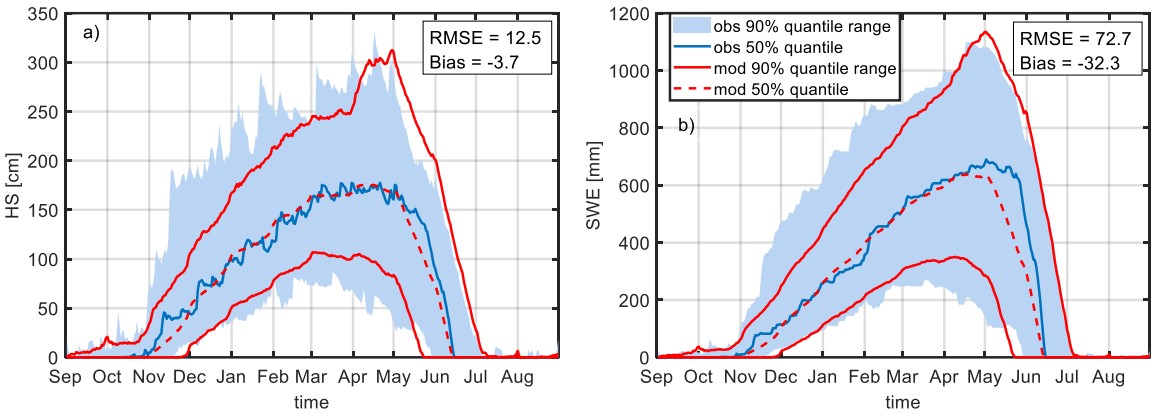

**Figure 5. Snow depth measured (left) and SWE (right) derived by an HS2SWE model (blue) at station OBW 2. Plotted are the 5, 50 and 95 percentiles of 20 years of observations (obs) and 1500 years of weather-generator-snow-model data (mod). RMSE and bias are calculated for 50 percentile values and for days when either the model or the observations show positive values.**

## 3.2 Climate change impact

In this section, we first provide an overview of how natural climate variability and model uncertainty affect key inputs to snowpack modeling; second, we show projections of future seasonal SWE curves; third, we discuss changes in ROS properties; and finally, we provide a quantification of sources of uncertainty.

### 3.2.1 Natural climate variability and climate model uncertainty

Figure 6 shows the annual and spatial means of TA, precipitation, and SWE on April 1 for the current future climate conditions. Natural climate variability is shown with error bars, while climate model uncertainty can be interpreted with the differences between climate model chains. For TA, climate model uncertainty dominates, while for precipitation, natural climate variability dominates. This result is consistent with those presented in other studies (Hawkins and Sutton, 2009; 2011; Fatichi et al., 2016; Peleg et al., 2019). The total uncertainty range of SWE on April 1 is mainly generated by natural climate variability for the mid-century, while at the end of the century both sources of uncertainty contribute similarly. A more quantitative analysis of the specific uncertainty contributions can be found in Sect. 3.2.7. Note that all of the following figures show uncertainty ranges of climate period averages to illustrate how different equally likely realizations of a future climate period are. The inter-annual uncertainty range is much larger (not shown) and is not the subject of this paper.

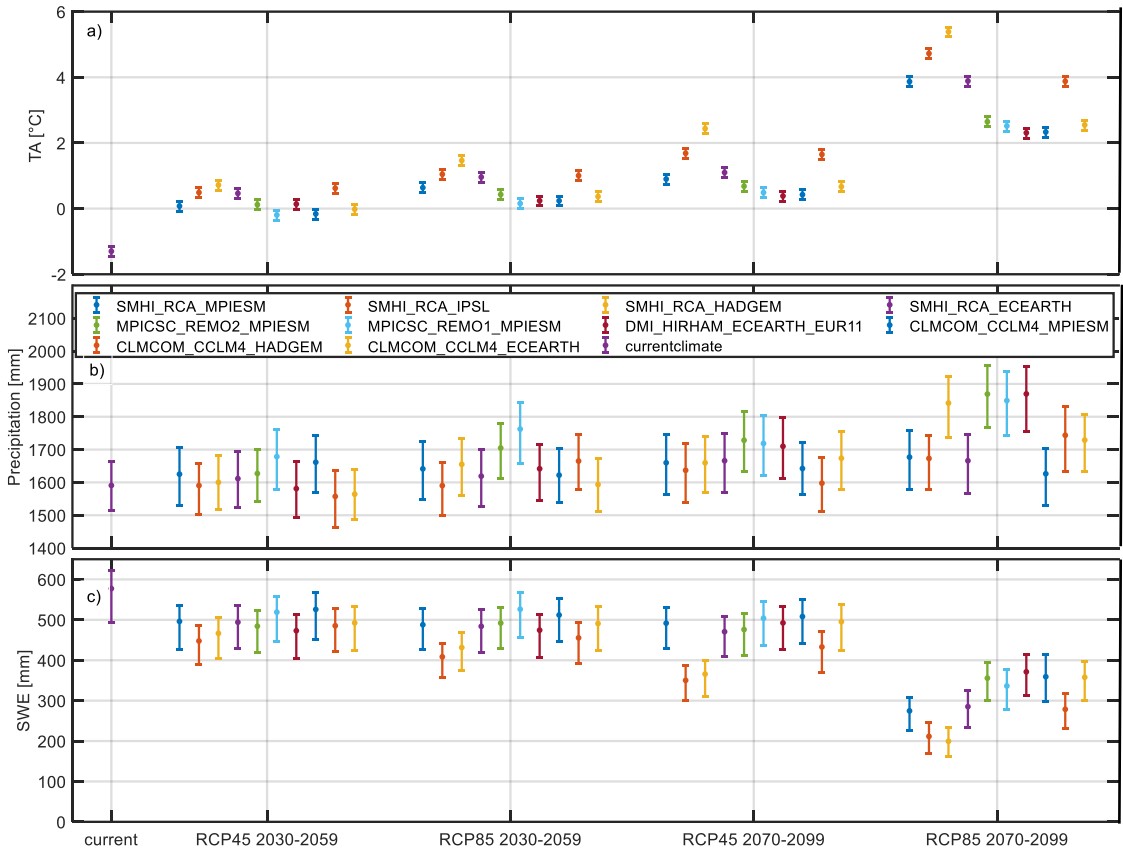

**Figure 6. Natural variability and climate model uncertainty of annual and spatial mean (a) precipitation, (b) air temperature (TA) and (c) resulting spatial mean SWE at April 1. Plotted are the 5, 50, 95 percentiles from 50 realizations of climate period mean values.**

### 3.2.2 Change in seasonal SWE

Figure 7 shows the seasonal evolution of areal mean SWE for different emission scenarios and periods. The uncertainty range for current climate (blue) is, by definition, only determined by natural variability, while for future climate (red) it is influenced by a composite of natural variability and climate model uncertainty. From May on, the changes in SWE for all emission scenarios and time horizons are larger than the uncertainty range (i.e. no overlap of uncertainty ranges). During the accumulation period, only the extreme emission scenario RCP 8.5 at the end of the century shows no overlap, while overlaps of up to 50% are achieved for the other cases. At the time of the SWE maximum in this region (May 1st), the overlap is already close to zero due to the onset of melting in the future scenarios. Similar to Verfaillie et al. (2018), the uncertainty in the emission scenarios is only relevant at the end of the century, as discussed in detail in Sect. 3.2.7.

For all scenarios, the altitude effects are similar. At the lowest altitudes (1400 - 1950 m a.s.l.), the climate signal is large enough to emerge clearly from the uncertainty ranges, while the largest overlap is achieved at the highest altitudes (3050 - 3600 m a.s.l.) (Fig. S1). Only for the most extreme scenario, RCP 8.5 at the end of the century, no overlap is achieved even at the

highest altitude range. This is generally consistent with the results of Marty et al. (2017), who also found a weakening of the climate change signal at higher elevations. Furthermore, the results are mostly consistent with Willibald et al. (2020) who found a similar elevation effect in how natural climate variability can mask trends in mean and maximum snow depth, although, the role of natural climate variability seems to be larger in their study than in our results. While at a low altitude site only 15%

of 50 realizations of future climate conditions under RCP 8.5 showed insignificant trends for time horizons until the mid of the century, at a high-altitude site (Weissfluhjoch, 2540 m) it was still 80%. For the latter station, they still found 20% of all realizations with insignificant trends until the end of the century. For our data for RCP 8.5 at the end of the century, no overlap is found for SWE for no time of the year and also not for the highest altitude range. Also, for low elevations at the mid of the century, hardly any overlap is exhibited (Fig. S1a).

In summary, these results suggest that the climate change signal for the area-averaged SWE is generally larger than the associated uncertainty. Only for elevations above 2000 m and for the months between January and April there are likely realizations of future climate with an equal amount of SWE as today. These exceptions can be characterized as situations where precipitation variability can strongly influence SWE amounts, i.e. when most of the precipitation falls as snow and melt is negligible. However, the later onset of SWE accumulation in future climate prevents natural variability from being able to

fully mask the climate change signal in the accumulation season, as is the case with precipitation (Peleg et al., 2019) or runoff (Fatichi et al., 2014, Moraga et al., 2021).

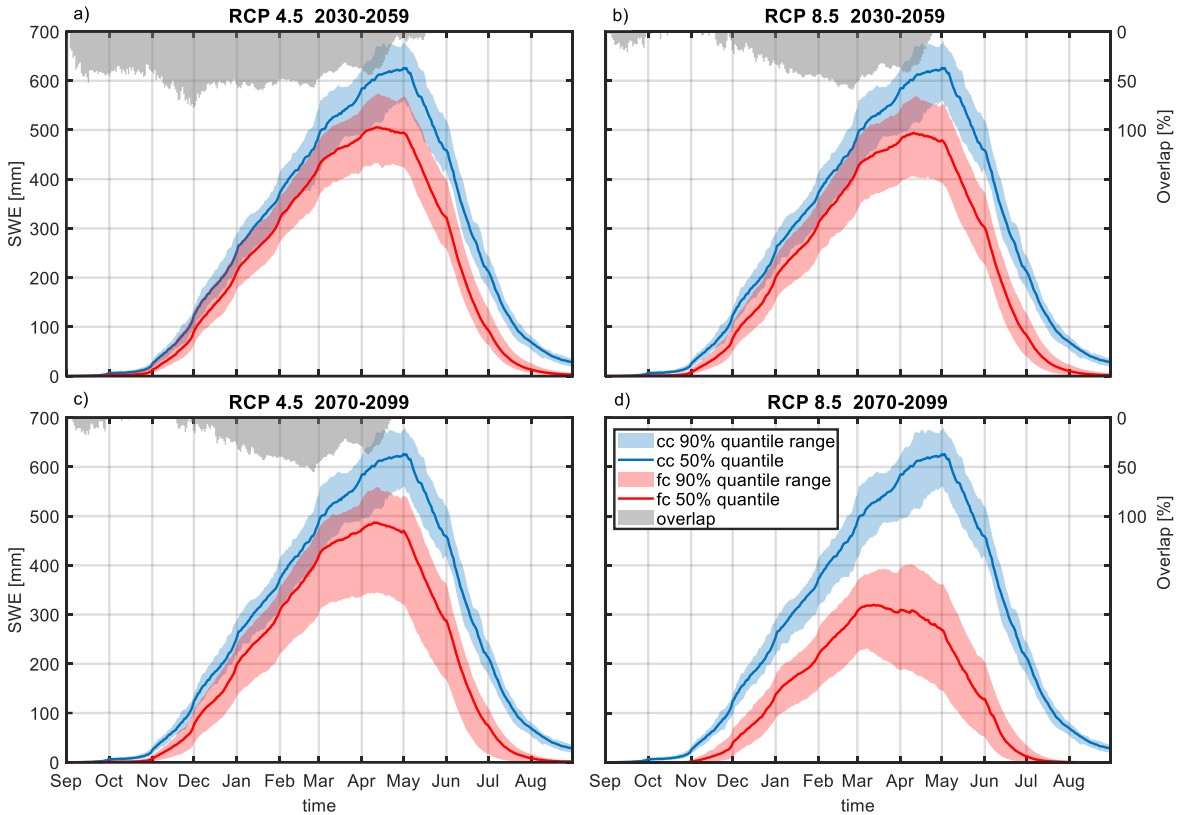

**Figure 7. Areal mean seasonal SWE development under current (cc) and future climate (fc) for different emission scenarios and time horizons. Plotted are the 5, 50, 95 percentiles of climate period mean values stemming from 50 (current climate) and 500 climate periods (future climate with 50 realizations of 10 climate models). The overlap indicates how much of the current climate natural variability is overlaid by the future climate uncertainty range.**

### 3.2.3 Frequency of rain-on-snow events

Figure 8 presents the exceedance probability of contributing area sizes of ROS events for all different pixel-based criteria (see Table 2) for RCP 8.5 at the end of the century. For example, in Fig. 8a, using criterion 1, approximately nine ROS events per year (exceedance probability of 0.0247) are simulated with a contributing area greater than 20% of the total area for current and future climate conditions. For this most extreme scenario, there is a climate change signal toward more frequent events for most of the contributing area size thresholds. However, whether or not the climate signal emerges from uncertainty ranges depends on the pixel-based criterion to define a ROS event. For criterion 4 and partially for criterion 2 (see Table 2) the signal of change is apparent, while it is not the case for criteria that also require 20% of the SWI contribution from snowmelt (criteria 1 and 3). Increasing the rainfall threshold results in a clearer climate change signal, likely because rainfall in higher precipitation intensities is more frequent at the end of the century (Fig. S3) due to more total precipitation (Fig. 6) and due to warmer air temperatures, which increase the liquid fraction. The reason why the increase in ROS frequency is masked when

the additional melt demand is used to define a ROS event can be found in the change of seasonality of ROS events and is
discussed in Sect. 3.2.4. For other emission scenarios and ROS definitions, the overlap is even more pronounced (Fig. S4).

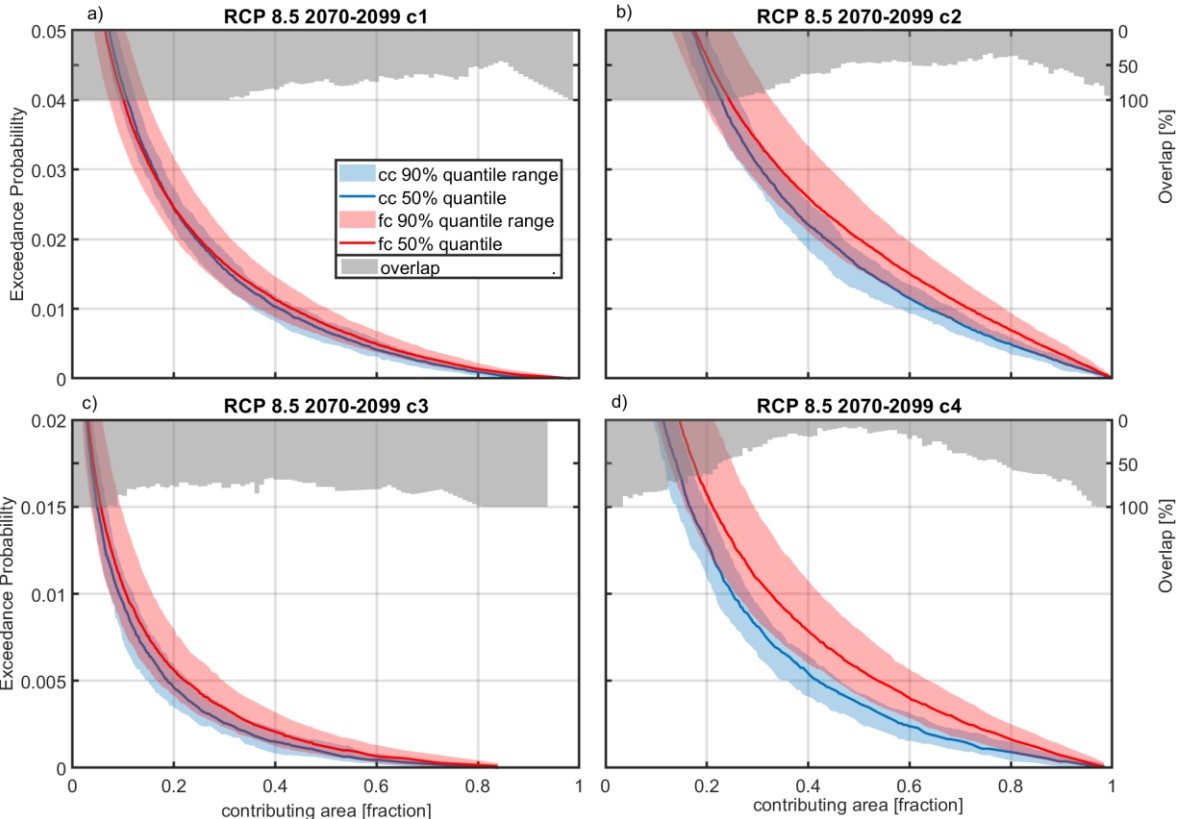

**Figure 8. Yearly exceedance probability of contributing area size (as a fraction of the total area) during ROS events for current climate (cc) and RCP 8.5 end of the century (fc) for criteria 1 (c1) to 4 (c4). Plotted are the 5, 50, 95 percentiles of climate period**
**mean values stemming from 50 (current climate) and 500 climate periods (future climate with 50 realizations of 10 climate models). The overlap indicates how much of the current climate natural variability is overlaid by the future climate uncertainty range.**

It is also worth noting the altitude dependence of this analysis for RCP 8.5 at the end of the century. At high elevations typically above 2500 m a.s.l., the increase in ROS events is pronounced for criteria 2 (not shown) and 4 (Fig. S5). For all other criteria defining ROS events and all other emission scenarios and periods, an increase at high altitudes above 2500 m a.s.l. is also
observed, but this is masked by the sources of uncertainty (e.g. Fig. S6 for criterion 1).

In summary, natural climate variability and climate model uncertainty question the claim that ROS events will become more frequent in a future climate in this high elevation study area, except for the most extreme scenario RCP 8.5 at the end of the century at high elevations above 2500 m if the ROS definition does not include a snowmelt contribution. If a ROS event is defined such that there must be a substantial snowmelt contribution (>20%), then a future increase in ROS frequency is masked
by the sources of uncertainty included in this study without any exceptions.

Thus, our results confirm our initial hypothesis that ROS events are strongly influenced by natural climate variability because they are more driven by precipitation than by seasonal SWE curves. However, some studies do find an increase in ROS frequency at higher elevations (e.g. Beniston and Stoffel, 2016; Musselman et al. 2018), and a discussion on this can be found in Sect. 3.2.6.

### 370    **3.2.4    Rain-on-snow seasonality**

In this section, we discuss why the climate change signal is more pronounced for frequencies of ROS events with only minor snowmelt contribution versus substantial contribution. Following the definition of criterion 2, each ROS event can spatially consist of pixels that will also satisfy criterion 1, i.e. with snowmelt contribution > 20% to SWI (see Fig. S7 for a spatial example). Figure 9 shows in a histogram the number of ROS events per month and their spatial characteristic computed as the

ratio $\varphi$ of pixels obeying criterion 1 over pixels obeying criterion 2. During current climate conditions (Fig. 9a), most ROS events occur from May to July, typically with large $\varphi$, i.e. a large spatial proportion of pixels with substantial snowmelt contribution. In January, for example, only a small number of ROS events occur, and most are characterized by a low $\varphi$, i.e. a small spatial proportion of pixels with substantial snowmelt contribution. This is consistent with the results of Würzer et al. (2016), who found that ROS events with a substantial snowmelt contribution typically occur in late spring and early summer,

when the snowpack is wet and warm at the onset of the event. Conversely, a low snowmelt contribution is expected when the initial snowpack is drier and colder (Würzer et al., 2016). Similar to Würzer et al. (2016), conditions for a substantial contribution from snowmelt are typically found under initially wet and warm snowpack conditions (Figs. S8a and b). This is indicated by the red arrow in Fig. 9, which points at large $\varphi$ that is associated with typically wet and warm initial snowpack conditions.

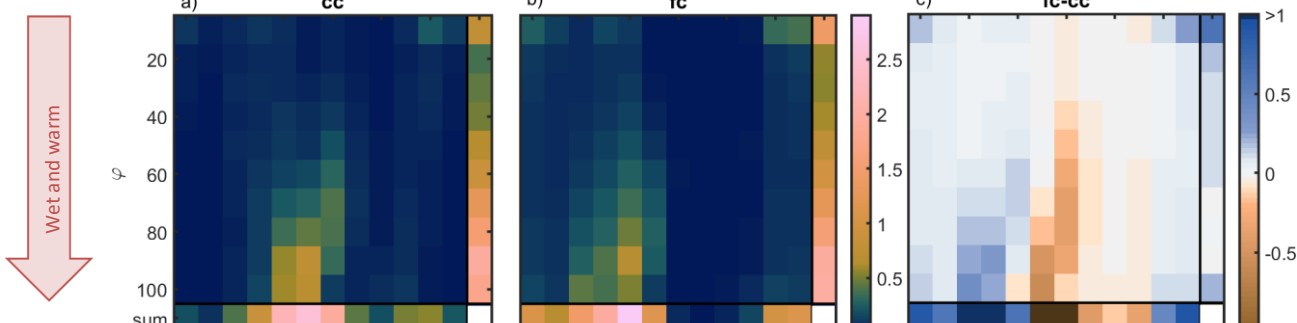

**Figure 9.** Histogram showing the number of ROS events for (a) current climate (cc) and (b) future climate (fc, RCP 8.5 at the end of the century), and (c) the difference thereof (fc-cc), split in different values of $\varphi$, i.e. the ratio of pixels obeying criterion 1 over pixels

obeying criterion 2 per event. The colour bar indicates the number of ROS events per year averaged over 1500 years for the current climate and 10 x 1500 years for future climate conditions. For the month with the lowest number of ROS events in the current climate, in February, there are approximately 100 ROS events available in the data set.

For RCP 8.5 at the end of the century (Fig. 9), the peak of ROS events shifts to earlier in the season, with typically large $\varphi$. There is also a higher number of early and midwinter events, with typically small spatial ratios, and almost no ROS events from July through September due to nonexistent snowpack.

The increase in ROS events in March and April with large $\varphi$ values contrasts with a large decrease in June and July (Fig. 9c). This means that the ROS events with spatially a large number of pixels with substantial snowmelt are not largely changing in the future with regards to their frequency, but rather shifted to earlier in the season. Only the frequency of events with small $\varphi$ will increase in the future. This may be seen as counterintuitive at first because warm and wet conditions are expected to occur more frequently in a future climate state. Indeed, this is generally the case at the onset of ROS events (Figs. S8c and d). However, rain in early and midwinter will fall on snow that will – even in this extreme warming scenario – be typically too cold and too dry to allow a significant contribution from snowmelt. This result implies that warmer air temperatures due to a changing climate can change the phase of precipitation more often than they can change the state of the snowpack to substantially contribute to runoff. This fact explains the limited increase in early and midwinter ROS events frequency with a large spatial proportion of substantially snowmelt contribution. Towards summer, the drastically reduced snow cover summer in a future climate explains the much faster decrease in the number of ROS events in this time, when ROS events have typically large $\varphi$, compared to the current climate.

In summary, the occurrence of rain falling on an initially warm and wet snowpack will likely not increase in the future. This explains that the climate change signal of ROS frequency shown in Figs. 8a and c are masked by uncertainty sources when a ROS event is defined by a substantial snowmelt contribution. However, a significant climate signal with varying signs is expected within individual months, e.g. March and June. These findings imply the need for a process-based snow model that can adequately model snowpack retention, as shown in this study.

### 3.2.5    Rain-on-snow intensity and snowmelt contribution

Since rain intensity is expected to increase significantly in a future climate for all scenarios studied, also during ROS conditions (Fig. S3), one can expect SWI to increase for rain-on-snow events as well. However, the conclusions are very similar to those for ROS frequency. An increase of high SWI intensities is observable but is masked by the sources of uncertainty quantified in this study for all emission scenarios and time horizons (see Fig. S9 for RCP 4.5 at the end of the century) except for the most extreme scenario (Fig. 10), i.e. RCP 8.5 at the end of the century, still depending, however, on the ROS definition criteria. If the ROS criterion implies a substantial contribution of snowmelt to SWI, again, the increase is masked by uncertainty, whereas without this condition this is not the case. The elevation dependence is also very similar to the ROS frequency (not shown): at higher elevations, the increase is pronounced for criteria 2 and 4 for elevations above 3000 m and 2500 m, respectively. For all other definitions of ROS events and all other emission scenarios and time horizons, this increase is also observed but is masked by sources of uncertainty.

Since snow cover decreases massively at the end of the century in the most extreme climate scenario RCP 8.5 (cp. Fig. 7d), it can be expected that the contribution of snowmelt to SWI also decreases and the observed increase in ROS events is mainly

driven by an increase in rain intensity. However, this depends on the pixel-based definition of whether a positive or negative climate signal can be observed. When substantial snowmelt contributions are required, the signal is largely masked by sources of uncertainty (Fig. S10). These results show that despite a dramatic decrease in snowpack by the end of the century in an RCP 8.5 scenario, the role of snow in contributing to runoff does not largely change for ROS events.

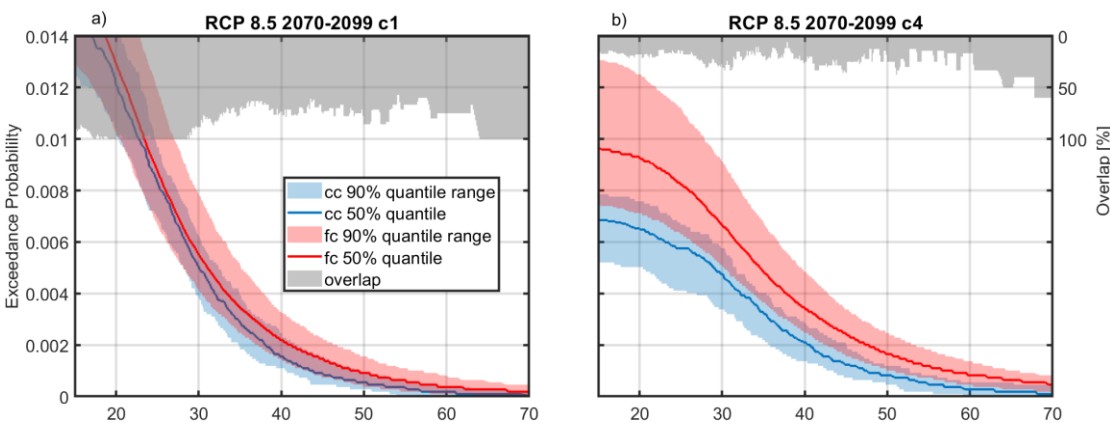

**Figure 10. Yearly exceedance probability of total area-averaged SWI of ROS events for current climate (cc) and RCP 8.5 at the end of the century (fc) for (a) criterion 1 (c1) and (b) criterion 4 (c4). A contributing area >1/3 of the total area was chosen to define a ROS event. Plotted are the 5, 50, 95 percentiles of climate period mean values stemming from 50 (current climate) and 500 climate periods (future climate with 50 realizations of 10 climate models).**

### 3.2.6    Comparison with other studies on ROS frequency

The results obtained here are based on a more complex approach than those of existing studies on this topic (e.g. Beniston and Stoffel, 2016; Morán-Tejeda et al., 2016; Musselman et al., 2018; Ohba and Kawese, 2020; Sezen et al., 2020), as we have added two new dimensions, i.e. internal climate variability and the ROS definition. Beniston and Stoffel (2016) reported that
in the Swiss Alps, an increase of nearly 50% in the number of ROS events occurred with 2-4 °C warmer temperatures than today at elevations of 2000 and 2500 m. For altitudes of 1500 m and below, a decrease in the number of ROS events was obtained. Except for two climate models, this temperature increase corresponds to the most extreme scenario RCP 8.5 at the end of the century (see Fig. 6a). Morán-Tejeda et al. (2016) came to very similar conclusions. Beniston and Stoffel (2016) and Morán-Tejeda et al. (2016) did use, however, empirical snow models without the capability that water retention can depend
on the state of the snowpack. Ohba and Kawase (2020) did not use snowmelt in their definition of ROS events and Sezen et al. (2020) defined ROS events with a very small amount of snowmelt (0.1 mm d$^{-1}$). Thus, reporting more ROS events at high elevations is consistent with our results using a criterion that does not imply a substantial snowmelt contribution. We claim, however, that the ROS definition must account for the runoff perspective and should not be based only on the occurrence of liquid precipitation on snowpack, because of the pronounced risk in flood potential due to excess runoff from snowmelt
(Würzer et al., 2006). Thus, it is important to note that our results using ROS definitions, which require a substantial snowmelt

contribution differ from existing studies, suggesting that more frequent rain on snow in the future does not result in a more frequent combination of rain and snowmelt, as highlighted in Section 3.2.4.

Musselman et al. (2018) defined ROS events identically to the criterion 1 chosen here (i.e. >10 mm rain per day, >10 mm SWE, and >20% snowmelt contribution to SWI). They analyzed spatial energy balance model runs on a 4-km grid in western
North America. Similar to the studies in Switzerland, they achieved a decrease in the number of ROS events at lower elevations and an increase at higher elevations for an RCP 8.5 emissions scenario by the end of the century, an increase of ROS intensity and a decrease in the contribution of snowmelt. These results can also be found in our study case, but they are largely obscured by sources of uncertainty (Figs. 10a and S10a). When natural climate variability is artificially suppressed in our analysis by plotting only the first realization of a climate period (Fig. S11; note that the first realizations of the current and future climates
are initialized with the same parameters in the weather generator), one can more clearly follow the conclusions of Musselman et al. (2018) of an increase in intensity and a decrease in snowmelt contribution.

The following two studies found a decrease in ROS events also in high elevated catchments: Chegwidden et al. (2020) found that ROS events as a cause of annual runoff maxima will disappear at lower elevations and slightly decrease at higher elevations by the end of the century. They discussed their differences to Musselman et al. (2018), who modelled a similar domain, with
465 having climate model differences and, mainly, analyzing only annual runoff maxima, while Musselman et al. (2018) analyzed all event magnitudes. In our study, we see an increase of ROS frequency independent of the event size for all except one ROS criterion (Fig. 8b-d). Cheggwidden et al. (2020) used energy balance-based hydrological models to investigate ROS events, thus encompassing as well the role of soil in changing high flows.

Sikorska-Senoner and Seibert (2020) analyzed historic observations and found a decreasing number of ROS events also in
high elevated catchments in Switzerland using a degree-day snow model with a fixed degree day factor and threshold temperature. The difference in findings can also be found in the ROS definitions. Sikorska-Senoner and Seibert (2020), used for example a quite small snowmelt threshold of 1 mm d$^{-1}$ which classifies alongside with other criteria a flood as a ROS event. Chegwidden et al. (2020), however, used the same ROS definition as Musselman et al. (2018), which is identical with criterion 1 in our study.

In summary, similar conclusions compared to the cited literature would be drawn if our approach were simplified, i.e. (i) one does not distinguish between substantial and non-substantial snowmelt contribution based on snowpack conditions and/or (ii) natural climate variability was not accounted for. This study shows that the inclusion of both natural climate variability and a snow model capable of modeling liquid water retention based on physical process representations provides new insights, particularly that only ROS events with no significant snowmelt contribution will occur more frequently in the future, while
ROS events with significant snowmelt contribution will mainly shift towards earlier in the year.

### 3.2.7 Uncertainty partitioning

Figure 11 shows the seasonal SWE ($\Delta$SWE) climate change signal and the partitioning of uncertainty into the individual sources for the middle and end of the century. Note that the individual sources in Figs. 11a and b are shown symmetrically

around the mean climate change signal for illustrative purposes only, and that the ratio is equal to the square root of the fractions shown in Figs. 11c and d. The climate change signal and also total uncertainty is the largest around May 1, which corresponds to the date when snow accumulation regularly ends under current climate conditions (cf. Fig. 7). In absolute terms (Fig. 11a and b), natural climate variability remains roughly the same between mid-century and the end of the century, which has also been noted by others (e.g. for air temperature projections by Yip et al., 2011).

The relative contributions can be assessed with Figs. 11c and d. At mid-century, natural climate variability is the dominant source of uncertainty, accounting for more than 50% during the main winter season. Climate model uncertainty is the second-largest source, while scenario uncertainty and model-scenario interaction account for only a few percentage points. This picture changes for the end of the century, where emissions scenario uncertainty is the main source, accounting for 40% to 60% during the main winter season. Climate model uncertainty is the second-largest source with a contribution of about 30%, followed by natural climate variability, whose contribution steadily decreases to just over 10% in May. At the beginning and end of the snow season, natural climate variability has a larger relative contribution than is normally observed during the season, which means that natural climate variability is particularly important for studies focusing on the duration of the snowpack. The increasing role of emissions scenario uncertainty in SWE projections towards the end of the century means that efforts to reduce uncertainties in snow projections should focus on limiting uncertainties associated with emissions scenarios, similar to efforts to improve climate models.

The larger role of scenario uncertainty at the end of the century was already visible in Fig. 7 and is mentioned by Verfaillie et al. (2018). Verfaillie et al. (2018) also quantified snow model uncertainty and concluded that physical snow modeling has a contribution of up to 20% of the simulated results after mid-century, which they considered secondary to climate model spread. It was mentioned that its influence on trends (or climate change signals) is likely much smaller but was not quantified more precisely. In this study, we were not able to quantify this additional source of uncertainty, but comparing these two studies, we can assume that natural climate variability and snow model uncertainty may be similar at the end of the century. This assumption needs to be proven by future studies that include all four types of uncertainty sources.

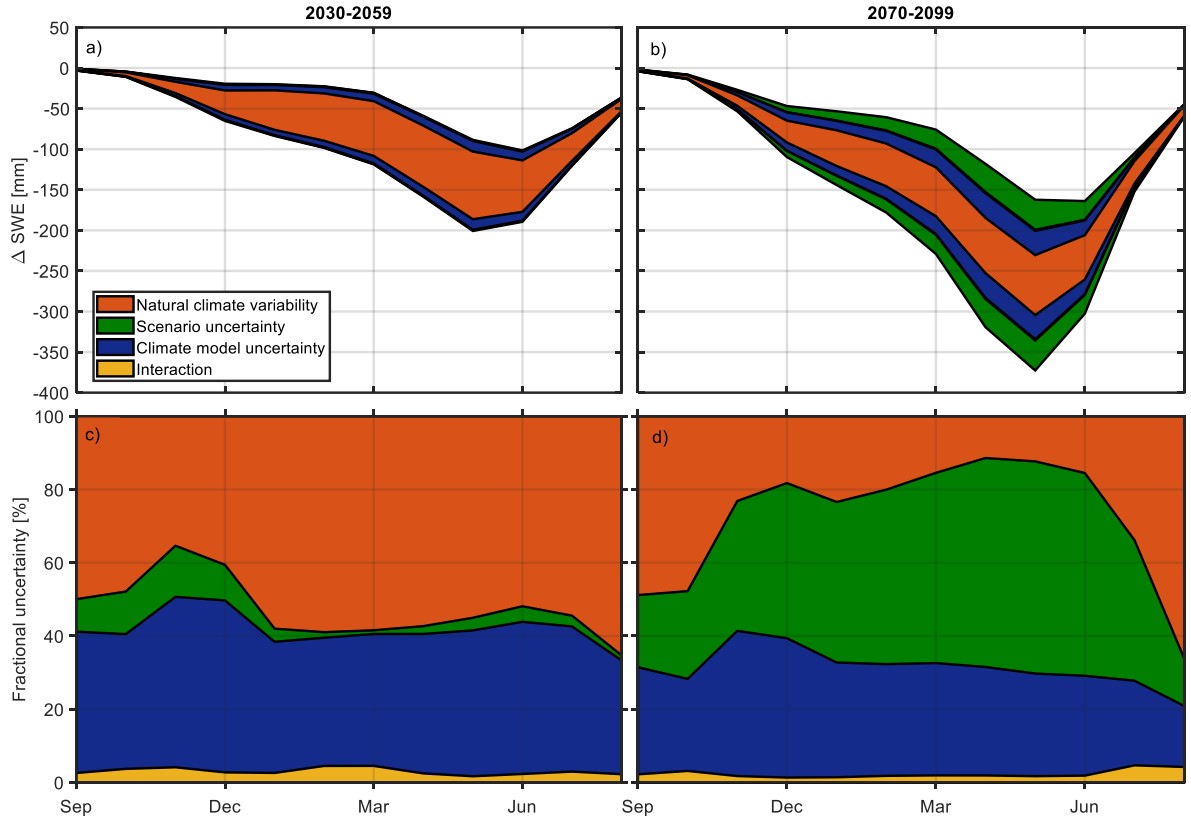

**Figure 11. Climate change signal of monthly mean SWE and illustration of the sources of uncertainty in the SWE projections (90% quantile ranges) of (a) mid and (b) end of the century. The fractional contribution of individual sources to the total uncertainty of (c) mid and (d) end of the century.**

Figure 12 shows the fractional contribution of uncertainty sources for the variables 'contributing area' and SWI determined with pixel-based criterion 3. Natural climate variability is the most dominant uncertainty source with increasing contributions for larger event sizes and larger runoff intensities with values larger than 70% of the total uncertainty range for event sizes larger than a third of the total area (Fig. 12b), or contributing-area averaged intensities larger than 20 mm/d (Fig. 12d), or of snow melt contributions larger than 30% (Fig. S12), even at the end of the century. These ratios depend on the pixel-based criterion, with the smallest contributions from natural climate variability obtained when using criterion 4, although still above 50% (Fig. S13). The larger contribution from the other sources of uncertainty may be explained by a clearer climate change signal for this criterion (cp. Fig. 8).

For the climate change signal of the ROS metrics studied here, natural climate variability is more important compared to ΔSWE, in agreement with our initial hypothesis, because the frequency of future ROS events depends more on precipitation and less on air temperature. Precipitation is more influenced by natural climate variability compared to air temperature at this spatial scale (Fatichi et al., 2016; Peleg et al., 2019). In fact, the relative contribution of the uncertainty sources of the ROS metrics studied here compares quite well on a local scale with the purely precipitation-based metrics in Fatichi et al. (2016).

This is in contrast to the continental scale studied in Hawkins and Sutton (2011), where the role of natural climate variability

in decadal mean precipitation diminishes and climate model uncertainty dominates toward the end of the century.

In summary, the total uncertainty in projections of the studied variables is composed of natural climate variability, climate model uncertainty, and emission scenario uncertainty, in this order for SWE projections only up to mid-century, and for all other variables up to the end of the century. The large contribution of natural climate variability demonstrates the need to quantify this source of uncertainty to prevent avoidable biases by end-users and decision-makers.

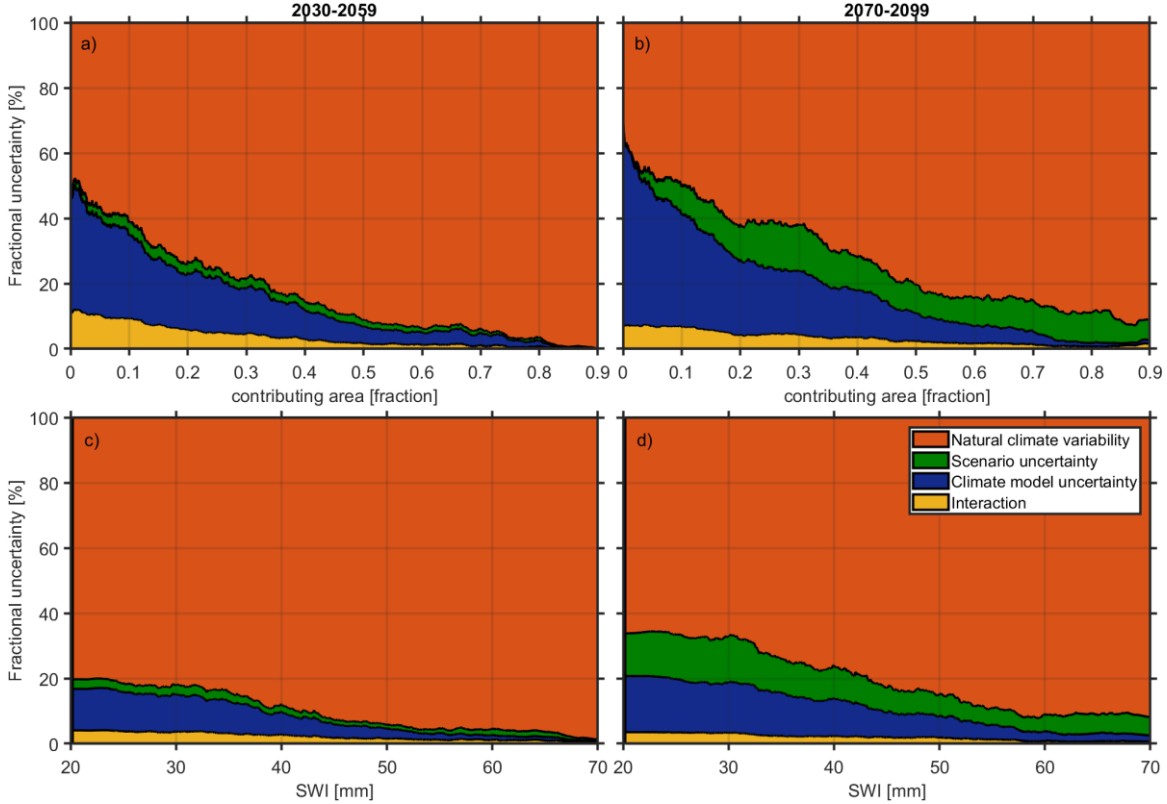

**Figure 12. Same as Fig. 11c and d for the exceedance probability of the variables 'contributing area' and contributing-area averaged SWI using criterion 3 (cp. Figs. 8 and 11).**

### 3.3    Limitations and generalizations

The weather generator AWE-GEN-2d is a hybrid approach that combines physical and statistical methods to derive climate variables, leading to intervariable dependence. Single Model Initial Condition Large Ensembles (SMILEs) (Maher et al., 2021) are alternatives to weather generators that quantify natural climate variability based solely on physical principles. However, for the use of studies similar to the one presented here, this method has significant disadvantages compared to weather generators. First, a SMILE depends on a single climate model with sometimes limited RCP availability (Lehner et al., 2020;

Maher et al., 2021), which does not allow to study the combined effect of natural climate variability, climate model uncertainty and scenario uncertainty. To overcome this problem, Lehner et al. (2020) used seven SMILEs and combined them with the CMIP5 and CMIP6 archives of the Coupled Model Intercomparison Project, which include multiple climate models but not multiple initial conditions, to distribute climate projection uncertainty. Willibald et al. (2020) downscaled a single SMILE with a single RCM for their assessment of natural climate variability of snow cover in the Swiss Alps, and thus were not able to

include the uncertainty of the emission scenarios and climate models as well. A third problem is the coarse spatial and temporal resolution; the resolution of the RCM SMILEs is in the order of 10 km (Maher et al. 2021). Willibald et al. (2020), for example, have downscaled, de-biased and disaggregated the RCM output to a sub-daily station scale using a univariate quantile mapping approach, which mitigates the initial advantage of benefiting from a purely physical variable interdependence in the climate model ensemble.

Besides the limitations in the physical description of the intervariable dependencies in AWE-GEN-2d, the large amount of data needed to train the model can be problematic, especially in ungauged areas; an alternative to using observed data can be the use of climate reanalysis data, as was demonstrated by Peleg et al. (2020). In addition, not all parameters in AWE-GEN-2d can be re-parameterized in the context of climate change. For example, we do not have the information of how to change the lapse rate of air temperature for future climate scenarios as the resolution of the physical climate models (e.g. RCMs) is

too coarse in space, which is certainly a limiting factor. But also empirical downscaling and debiasing methods like the widely used quantile mapping approach suffer from similar limitations. Another limiting point is that typical temporal dependencies in the data, e.g. due to synoptic patterns in a region, cannot be mapped in AWE-GEN-2d. Heavy winter precipitation can be related to cold frontal passages in certain regions, which can lead to a correlation between low temperatures and high precipitation intensity. This can have a significant impact on the precipitation phase and the resulting snow cover. It is

questionable whether relatively coarse scaled RCMs can model these dependencies in complex regions like the Alps. Moreover, if data are needed at a sub-daily and local scale, these dependencies may be lost with univariate downscaling routines. In summary, we think that a two-dimensional weather generator is a good alternative to using multiple SMILEs in combination with a downscaling routine when the complete chain of uncertainties is needed together with a very high (sub-kilometer and sub-daily) resolution. Note that the weather generator is only capable of detecting frequencies of natural

variability on the order of the training period (i.e. 30 years). Lower frequencies however, which may arise from processes

within the coupled ocean-atmosphere system via dynamic and thermodynamic interactions (Deser et al., 2012b), cannot be detected. Therefore, the relative contribution of natural climate variability might be underestimated in this study.

The transferability of the results to other areas found in the limited extent of our study area is complex. ROS events depend on a non-trivial interaction of the spatial distribution of liquid precipitation and the existing snow cover and its condition. The transferability to other regions is limited, as precipitation and temperature dependencies differ strongly from mountain region to mountain region. The different dependence between air temperature and short-wave radiation in mountain regions at other latitudes will also limit transferability. However, we assume that the dominance of natural climate variability to total uncertainty remains at this spatial scale also in other areas. We, therefore, believe that the well-described increase in ROS frequency due to a changing climate in high altitude areas from the western US to Europe and Japan is questioned with this study. This study motivates making such results more robust by quantifying natural climate variability.

## 4    Conclusions

The climate change signal of snow water resources and of ROS frequency and intensity was investigated with their climatic uncertainties. For the exemplary selected high-altitude study area in the Swiss Alps, the climate change signal towards fewer snow water resources during the ablation period was found to emerge clearly from the sources of uncertainty for all scenarios investigated. However, given significant uncertainties, there is some overlap during the accumulation period for all but the most extreme scenario (RCP 8.5, end of the century).

For ROS events, previous studies have shown that they will become more frequent and intense at higher elevations due to a shift toward liquid precipitation and despite a decreasing snowpack. The additional inclusion of natural climate variability in the uncertainty assessment revealed that this source is responsible for 70-90% of the overall uncertainty, similar to purely precipitation-based metrics. As a result, for all scenarios, including RCP 8.5 at the end of the century, the climate change signal of ROS frequency and intensity is larger than the uncertainty range only for events with no significant contribution of snowmelt to runoff (<20%). For events with a significant contribution of snowmelt to runoff, the climate change signal is too small and could potentially only be explained by natural climate variability. These events regularly occur during conditions with an initial warm and wet snowpack. The very rapid decline in snowpack toward early summer in future climate, when conditions typically prevail for substantial contributions from snowmelt, will result in a large decrease in such ROS events that cannot be compensated for at other times of the year: in early and midwinter, when rain is expected to fall more often in a future climate, it will fall on snow that will be typically too cold and too dry to allow a significant contribution from snowmelt. Warmer air temperatures due to a changing climate are more likely to change the phase of precipitation than the condition of the snowpack to contribute significantly to runoff. This implies that ROS events with a significant contribution of snowmelt to runoff will occur earlier in the year, but not more frequent under future climate.

These additional results were possible only with increased model complexity, first by using a snow model that represents water retention in snow based on physical processes, and second by accounting for natural climate variability to quantify the signal-

to-noise ratio of climate at the local scale. Natural climate variability, climate model uncertainty, and emission scenario uncertainty, in this order, composed the total uncertainty for SWE projections up to mid-century, and for ROS projections up to the end of the century. Therefore, it is vital to quantify natural climate variability in snow projections to avoid bias among end-users and decision-makers.

## 5    Appendix

Factors of change (FC) were calculated following Peleg et al. (2019) with

$$P^{FUT} = \frac{P^{CLM,FUT}}{P^{CLM,CUR}} P^{OBS} \tag{1}$$

$$TA^{FUT} = TA^{OBS} + (TA^{CLM,FUT} - TA^{CLM,CUR}) \tag{2}$$

where $P$ stands for precipitation variables (mean and variance of precipitation intensity and precipitation occurance) and $TA$ for mean air temperature, $FUT$ and $CUR$ denote future and current climate realizations (respectively), CLM denotes the climate model and OBS denotes the observed data.

To obtain the relative contribution of the investigated sources of uncertainty (i.e. natural climate variability $I$, climate model uncertainty $M$, and emission scenario uncertainty $S$) to the total uncertainty $T$, the partitioning method presented by Yip et al. (2011) was applied, following Eq. (3) – (7):

$$V(t) = \frac{1}{N_m N_s N_r} \sum_{m=1}^{N_m} \sum_{s=1}^{N_s} \sum_{r=1}^{N_r} [x(m,s,r,t) - x(m,s,\cdot,t)]^2 \tag{3}$$

$$M(t) = \frac{1}{N_m} \sum_{m=1}^{N_m} [x(m,\cdot,\cdot,t) - x(\cdot,\cdot,\cdot,t)]^2 \tag{4}$$

$$S(t) = \frac{1}{N_s} \sum_{s=1}^{N_s} [x(\cdot,s,\cdot,t) - x(\cdot,\cdot,\cdot,t)]^2 \tag{5}$$

$$I(t) = \frac{1}{N_m N_s} \sum_{m=1}^{N_m} \sum_{s=1}^{N_s} [x(m,s,\cdot,t) + x(\cdot,\cdot,\cdot,t) - x(m,\cdot,\cdot,t) - x(\cdot,s,\cdot,t)]^2 \tag{6}$$

$$T(t) = \frac{1}{N_m N_s N_r} \sum_{m=1}^{N_m} \sum_{s=1}^{N_s} \sum_{r=1}^{N_r} [x(m,s,r,t) - x(\cdot,\cdot,\cdot,t)]^2 = V(t) + M(t) + S(t) + I(t) \tag{7}$$

where $x(m,s,r,t)$ is a climate period mean climate change signal for model $m$, scenario $s$, replication $r$ and time horizon $t$, $x(\cdot,\cdot,\cdot,t)$ the overall mean at time horizon $t$, $x(m,s,\cdot,t)$ is the mean over all replications, $x(m,\cdot,\cdot,t)$ and $x(\cdot,s,\cdot,t)$ is the mean over the scenarios and replicates, and the mean over the models and replicates, respectively. Replications are the $N_r = 50$ realizations of a climate period, $N_m = 10$ for the ten climate model chains, and $N_s = 2$ for the two emission scenarios. The

interaction term takes into account that climate model uncertainty and emission scenario uncertainty might be correlated, e.g.
that the warmest model for RCP 4.5 does not need to be the warmest in RCP 8.5.

Fractional uncertainties were calculated scaling each individual source with the total uncertainty. Additionally, we followed the method of Hawkins and Sutton (2011) and Lehner et al. (2020) to obtain 90% quantile ranges of uncertainty sources, assuming symmetry around the overall mean $x(\cdot,\cdot,\cdot,t)$:

$$Q_V(t) = x(\cdot,\cdot,\cdot,t) \pm 1.645 \frac{\sqrt{V}}{F}, \tag{8}$$

$$Q_{V+M}(t) = x(\cdot,\cdot,\cdot,t) \pm 1.645 \frac{\sqrt{V} + \sqrt{M}}{F}, \tag{9}$$

$$Q_{V+M+S}(t) = x(\cdot,\cdot,\cdot,t) \pm 1.645 \frac{\sqrt{V} + \sqrt{M} + \sqrt{S}}{F}, \tag{10}$$

$$Q_{V+M+S}(t) = x(\cdot,\cdot,\cdot,t) \pm 1.645 \frac{\sqrt{V} + \sqrt{M} + \sqrt{S}}{F}, \tag{11}$$

$$Q_{T(t)} = Q_{V+M+S+I}(t) = x(\cdot,\cdot,\cdot,t) \pm 1.645 \frac{\sqrt{V} + \sqrt{M} + \sqrt{S} + \sqrt{I}}{F}, where \tag{12}$$

$$F = \frac{\sqrt{V} + \sqrt{M} + \sqrt{S} + \sqrt{I}}{\sqrt{V + M + S + I}}. \tag{13}$$

Since the assumption of symmetry does not necessarily hold, the corresponding figures (Fig. 11a and b) are for illustrative
purposes only.

## 6    Data availability

Daily data of simulated current and future climate periods are available at (https://www.envidat.ch/#/metadata/multiple-realizations-of-daily-swe-swi-and-rain-projections). This is a preliminary dataset, as the review process of the associated manuscript has not yet been completed. Once the final version of the manuscript is published, the dataset will be assigned a
DOI.

## 7    Author contribution

MS led the project, created and verified the modeled data set, analyzed the data, and wrote the manuscript. NP set up and trained the weather generator, verified the modeled data and discussed the results. AW worked on the snow model and discussed the results. TJ provided ideas for data analysis and discussed results. TJ, PB and NP contributed to the writing and
editing of the manuscript.

## 8    Competing interests

The authors declare that they have no conflict of interest.

## 9    Acknowledgments

MS would like to thank Thomas Kramer for HPC computing support and Louis Queno and Nora Helbig for the continued
development of the snow model. He would also like to thank Massimiliano Zappa for providing funding to complete the
manuscript. MS and NP were partly funded by the Swiss Competence Center for Energy Research-Supply of Electricity
(http://www.sccer-soe.ch).

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
