# Peer review of "Natural climate variability is an important aspect of future projections of snow water resources and rain-on-snow events"

_The Cryosphere, 2021_

## Referee Comment (RC1)

**Review for manuscript "Natural climate variability is an important aspect of future projections of snow water resources and rain-on-snow events"**

**Authors:** Michael Schirmer, Adam Winstral, Tobias Jonas, Paolo Burlando, and Nadav Peleg

**Journal:** The Cryosphere

**Summary**

The authors use a model chain consisting of climate models, a weather generator, and an energy balance snow model to identify dominant uncertainty sources in future changes in snow-water-equivalent and rain-on-snow runoff. They show that changes in ROS events emerge till the end of the century despite large uncertainties while ROS events with substantial snowmelt contributions don't show a clear change signal.

**General remarks**

The study by Schirmer et al. builds on a complex model chain consisting of climate models, a weather generator and an energy balance snow model to assess the importance of internal variability on the detection of future changes in snow and rain-on-snow runoff events. I think that the combination of different model types to better describe internal variability is a generally a valid approach to determine the importance of internal variability in change assessments of snow-related quantities compared to other uncertainty sources. However, I see a substantial need for clarification regarding the research questions and methodology and think that the approach chosen to decompose uncertainty into different contributors needs refinement. Given the current 'incomplete' methods descriptions, it is difficult to assess the validity of the results. Furthermore, I think that the manuscript would profit from reorganization, i.e. restructuring the methods section following a more logical sequence and from separating the results from the discussion. Finally, the manuscript would in my opinion profit from a visualization of the most important modeling steps and their relationships and from refining figures by adapting color schemes and adding legends. Please find my more detailed review below.

**Major points**

1. Research questions: the research questions are not entirely clear and should be explicitly stated in the introduction. From how I understand the study it is something along the lines of: 'How does the importance of internal variability differ between temperature-driven snow resources and rain-driven rain-on-snow events' and 'When is the time of emergence of changes in snow availability and rain-on-snow runoff.'
2. Introduction: In addition to model-based studies looking at changes in rain-on-snow floods, there are also observation-based studies, which I think should be mentioned in the introduction. E.g. Sikorska and Seibert (2020; 10.1080/02626667.2020.1749761) or Cheggwidden et al. (2020; 10.1088/1748-9326/ab986f).
3. Methods section organization: The methods section does not seem to follow a logical order and could in my opinion be more logically organized by following a 'chronological' modeling order. E.g. Area, Climate models, Weather generator, Snow model, Rain-onsnow definition, Change assessment, Uncertainty decomposition. Providing a flowchart linking the most important modeling and analysis steps might enable further improvements in communication. Furthermore, the methods section lacks important methodological detail, which makes it difficult to assess the validity of the results.

4. Glacier retreat: The study region is influenced by a glacier, which affects runoff formation. However, the glacier-related changes in flow are not represented in the modeling chain (l. 86-87). This does not seem to be justified and might explain why melt-influenced changes in ROS events are don't show up clearly.

5. Weather generator: The weather generator description (Section 2.3) lacks important detail and it is therefore difficult to assess the validity of the approach. E.g. how does the weather generator use the climate simulations, how does the weather generator work, how is the temporal downscaling performed (l. 122), how are the different variables generated (l. 122), how is the inter-variable consistence (l.124) evaluated?

6. Snow model and variables: It remains unclear to me how the weather generator output is used to derive different snow-related variables (Section 2.2). Was the analysis performed per grid cell? Which variables were exactly derived? How was the model calibrated (l.95)? And what does the 'unpublished' model adjustment (l. 95-96) do?

7. Bias correction: Section 2.5. suggests that some bias correction might have been necessary to adjust simulated to observed values. Was such bias correction performed and if so why?

8. Uncertainty partitioning: The uncertainty partitioning procedure described in Section 2.6. does not seem to properly separate internal variability (residuals) from the signal. Or at least I can not see how the different uncertainty components have been decomposed e.g. using a procedure such as the one proposed by Hawkins and Sutton (2009; 10.1175/2009BAMS2607.1). The procedure used to derive climate model uncertainty also seems to encompass internal variability (l.153-154) and the procedure used to derive internal variability also seems to include climate model uncertainty (l.156-157). Furthermore, it would be nice to compute fractional uncertainty contributions that add up to 1, which currently does not seem to be the case.

9. Validation: I think that the methods section needs a 'Validation' subsection describing how the different models were evaluated. E.g. how were the validation stations chosen? Which variables were validated, …

10. Rain-on-snow events: how have these events been defined? There is a section called 'rain-on-snow' definition, which does, however, not really explain what you understand by a 'rain-on-snow' event. How is the 'surface water input' computed?

11. Results: I would clearly separate the results part from the methods section and discussion. Some parts can be moved from the Results to the Methods section (e.g. l. 206-216) and other parts to a newly created Discussion section (essentially everything that compares the study's findings to findings of existing studies). Furthermore, it would be nice if the results section followed a similar structure as the methods section.

12. Figure 11: Would be nice to depict the fractional contributions of the individual uncertainty sources. Currently, all of the sources seem to not be clearly separated (probably also related to the issue risen in comment 8).

13. Discussion: I miss a discussion of alternative strategies that could be used to quantify the relative importance of internal variability to other uncertainty sources besides weather generators. Recently, quite a few studies have used single model initialized large

ensembles (SMILES) for uncertainty decomposition and I think it would be important to mention this (e.g. Maher et al. 2021; 10.5194/esd-12-401-2021 or Lehner et al. 2020; 10.5194/esd-11-491-2020). Other topics I would address in the discussion section include: model deficiencies, comparisons of results with other studies (as distributed throughout the results section), and the generalizability of the findings to other geographical and climatic contexts given that the study relies on one catchment only.

14. Figures: Figure design should be improved by using continuous color schemes for continuous variables (e.g. Figure 1c) and by avoiding the use of rainbow color schemes, which are not color-blindness friendly (Figure 8a and 8b). Furthermore, complete legends should be provided for all figures.

**Minor points**

- L. l. 40-41: I don't think that it is correct to say that 'no studies have previously included internal climate variability in their analysis' as internal climate variability is per definition part of every change impact assessment. However, it might be ok to say that 'the relative importance of internal variability compared to other uncertainty sources has not been previously assessed.'
- L.58-59: needs rephrasing.
- L. 61: 'they' will
- L. 63: that 'drive runoff'
- L. 72-73: repeats content already provided in l. 68-69 and can be removed.
- L. 140: How do you define 'frequency', number of events per year or non-exceedance probability or anything else?
- L. 259-260: Not sure whether the statement that 'the uncertainty range for current climate is, by definition, only determined by natural variability' is true. If the simulations for the current period are run with different climate models (which is as I understand it the case), climate model uncertainty might also be present.
- L. 279-280: Don't understand this sentence and think it needs rephrasing.
- L. 293: Sentence seems incomplete. Would be nice to repeat the pixel based criteria, which the reader might no longer have present at this stage.

---

## Author Comment (AC1)

**Reviewer 1**

*We want to thank Reviewer 1 for her/his constructive comments. Our answers are written in italic letters alongside to the reviewer's comments.*

**Summary**

The authors use a model chain consisting of climate models, a weather generator, and an energy balance snow model to identify dominant uncertainty sources in future changes in snow-water-equivalent and rain-on-snow runoff. They show that changes in ROS events emerge till the end of the century despite large uncertainties while ROS events with substantial snowmelt contributions don't show a clear change signal.

**General remarks**

The study by Schirmer et al. builds on a complex model chain consisting of climate models, a weather generator and an energy balance snow model to assess the importance of internal variability on the detection of future changes in snow and rain-on-snow runoff events. I think that the combination of different model types to better describe internal variability is a generally a valid approach to determine the importance of internal variability in change assessments of snow-related quantities compared to other uncertainty sources. However, I see a substantial need for clarification regarding the research questions and methodology and think that the approach chosen to decompose uncertainty into different contributors needs refinement. Given the current 'incomplete' methods descriptions, it is difficult to assess the validity of the results. Furthermore, I think that the manuscript would profit from reorganization, i.e. restructuring the methods section following a more logical sequence and from separating the results from the discussion. Finally, the manuscript would in my opinion profit from a visualization of the most important modeling steps and their relationships and from refining figures by adapting color schemes and adding legends. Please find my more detailed review below.

*Thank you for your assessment. We agree that the method description needs to be expanded and that the manuscript would benefit from a modelling flowchart, a more chronological sequence with a clearer separation of sections, and a refinement of the figures. For more details, please see our specific responses to your comments below.*

**Major points**

1. Research questions: the research questions are not entirely clear and should be explicitly stated in the introduction. From how I understand the study it is something along the lines of: 'How does the importance of internal variability differ between temperature-driven snow resources and rain-driven rain-on-snow events' and 'When is the time of emergence of changes in snow availability and rain-on-snow runoff.'

*We will rework the final paragraph of the introduction to ensure that the research questions are concisely formulated.*

2. Introduction: In addition to model-based studies looking at changes in rain-on-snow floods, there are also observation-based studies, which I think should be mentioned in the introduction. E.g. Sikorska and Seibert (2020; 10.1080/02626667.2020.1749761) or Cheggwidden et al. (2020; 10.1088/1748-9326/ab986f).

*We agree and add these studies in the introduction.*

3. Methods section organization: The methods section does not seem to follow a logical order and could in my opinion be more logically organized by following a 'chronological' modeling order. E.g. Area, Climate models, Weather generator, Snow model, Rain-on-snow definition, Change assessment, Uncertainty decomposition.

*Our current methods section is structured so that the most important methods for this manuscript are described first (i.e. the weather generator and the snow model). However, we can certainly put the methods in a more chronological order, as suggested, to clarify the methodological flow.*

Providing a flowchart linking the most important modeling and analysis steps might enable further improvements in communication. Furthermore, the methods section lacks important methodological detail, which makes it difficult to assess the validity of the results.

*We agree that adding a flowchart as well providing more details in the methods section will improve the manuscript, which we will implement as suggested.*

4. Glacier retreat: The study region is influenced by a glacier, which affects runoff formation. However, the glacier-related changes in flow are not represented in the modeling chain (l. 86-87). This does not seem to be justified and might explain why melt-influenced changes in ROS events are don't show up clearly.

*Yes, this study region is influenced by a large glacier. However, we do not want to investigate the combined effects of snow and glacier retreat on mean snow water resources or ROS properties, but only to show the effects of snow. For this reason, we have not compared observed runoff measurements in the area with our modelled data. We chose this catchment because of its good data availability and high elevation. The study area is treated as if there were no glacier. Only the altitudes and the meteorological input data play a role for the study area. It can therefore be considered rather as an example area, perhaps even a virtual area, for which the influence of natural climate variability on snow water resources and ROS events is studied as an example. These considerations will be presented in more detail in the next version of the manuscript.*

5. Weather generator: The weather generator description (Section 2.3) lacks important detail and it is therefore difficult to assess the validity of the approach. E.g. how does the weather generator use the climate simulations, how does the weather generator work, how is the temporal downscaling performed (l. 122), how are the different variables generated (l. 122), how is the inter-variable consistence (l.124) evaluated?

*There are two comprehensive publications, Peleg et al. (2017, 2019), which we cite in this manuscript to answer these questions. We understand that a manuscript should stand on its own, but we also had to consider the readability of this already quite long manuscript. This is especially true since this study is an application to the weather generator among other methods. We have also not gone into detail on energy balance snow modelling or climate modelling. Given the complexity of, for example, how the weather generator works in detail, we have chosen to provide references to these issues, but at the same time to show in broad terms how the weather generator was trained and applied. We make sure that the references for these important questions are clearly visible, but to address you concern will also include more details from cited publications.*

6. Snow model and variables: It remains unclear to me how the weather generator output is used to derive different snow-related variables (Section 2.2). Was the analysis performed per grid cell? Which variables were exactly derived? How was the model calibrated (l.95)? And what does the 'unpublished' model adjustment (l. 95-96) do?

*In the more chronological order of the methods section suggested above we will include more details answering these questions. For now, the results achieved with the snow model are mean values of grid cells, either within the whole catchment or within all ROS affected pixel. This is described in each results section and will be placed also in the method section.*

*The model is an open-source energy balance snow model, which is publicly available. These types of models are usually not calibrated, and we did not calibrate our model as well. We will also provide more details on the adjustments to the model.*

7. Bias correction: Section 2.5. suggests that some bias correction might have been necessary to adjust simulated to observed values. Was such bias correction performed and if so why?

*Yes, the climate model chains were bias corrected and downscaled with quantile mapping (see section 2.4). We will state this more clearly, and also why we have chosen to use bias corrected climate model output.*

8. Uncertainty partitioning: The uncertainty partitioning procedure described in Section 2.6. does not seem to properly separate internal variability (residuals) from the signal. Or at least I can not see how the different uncertainty components have been decomposed e.g. using a procedure such as the one proposed by Hawkins and Sutton (2009; 10.1175/2009BAMS2607.1). The procedure used to derive climate model uncertainty also seems to encompass internal variability (l.153-154) and the procedure used to derive internal variability also seems to include climate model uncertainty (l.156-157). Furthermore, it would be nice to compute fractional uncertainty contributions that add up to 1, which currently does not seem to be the case.

*We follow the uncertainty partition method described by Fatichi et al. (2016), which is significantly different from other partition methods (such as the one described by Hawkins and Sutton that the reviewer mentioned). In our case, the reason to use the first method and not the second is that it is more suitable for the situation where you have the complete internal climate variability - model uncertainty - scenario uncertainty chain (as obtained from the AWE-GEN-2d model). Using Fatichi et al.'s method, the fractional uncertainty contribution cannot be summed to 1 (see their paper for more details). In light of the reviewer comment (and a similar comment given by the second reviewer), we see the need to explain the method and its differences from other uncertainty partition methods in more detail. Accordingly, the relevant section of the manuscript will be extended.*

9. Validation: I think that the methods section needs a 'Validation' subsection describing how the different models were evaluated. E.g. how were the validation stations chosen? Which variables were validated, …

*We will add a validation subsection in the methods section in order to be clearer on this topic.*

10. Rain-on-snow events: how have these events been defined? There is a section called 'rain-on-snow' definition, which does, however, not really explain what you understand by a 'rain-on-snow' event. How is the 'surface water input' computed?

*We do not define ROS events, but apply daily pixel-based criteria that then result in a certain size of "contributing area" each day. An "ROS day" can then be defined as a day with a contributing area of a certain size, which may depend on the application or the user.  We determined a climate change signal of ROS frequency with variable event sizes (x-axis of Figure 7). The intensity and contribution of snowmelt required a certain size of event (> 1/3 of the total area affected according to pixel-based criteria).  Since the reviewers have pointed out that methods and results should be more clearly separated, we will better organize these details and expand this topic.*

*The "surface water input" is calculated with the energy balance snow model and is the water input available at the ground surface through either snowpack runoff, or rain in case of snow-free conditions, or a mixture of both in case of fractional snowcover. We will add this definition in the methods section.*

11. Results: I would clearly separate the results part from the methods section and discussion. Some parts can be moved from the Results to the Methods section (e.g. l. 206-216) and other parts to a newly created Discussion section (essentially everything that compares the study's findings to findings of existing studies). Furthermore, it would be nice if the results section followed a similar structure as the methods section.

*As already written above, we will add more details in the methods section, which will also help to separate results from methods. We thank the reviewer for this suggestion.*

12. Figure 11: Would be nice to depict the fractional contributions of the individual uncertainty sources. Currently, all of the sources seem to not be clearly separated (probably also related to the issue risen in comment 8).

*Yearly mean fractional contributions are summarized in Figure 12. We think that including all daily fractional uncertainties also in Figure 11 would look to busy.*

13. Discussion: I miss a discussion of alternative strategies that could be used to quantify the relative importance of internal variability to other uncertainty sources besides weather generators. Recently, quite a few studies have used single model initialized large ensembles (SMILES) for uncertainty decomposition and I think it would be important to mention this (e.g. Maher et al. 2021; 10.5194/esd-12-401-2021 or Lehner et al. 2020; 10.5194/esd-11-491-2020). Other topics I would address in the discussion section include: model deficiencies, comparisons of results with other studies (as distributed throughout the results section), and the generalizability of the findings to other geographical and climatic contexts given that the study relies on one catchment only.

*We will add a discussion on other ways to quantify natural climate variability, along with the suggested publications. We note, however, that ensembles derived from other climate models (SMILES, for example) have much coarser spatial and temporal resolutions than ours. Finding other studies that partition the uncertainties for the processes we are simulating at fine scales is therefore not easy. The findings will also be discussed with regard to model deficiencies and generalizability.*

14. Figures: Figure design should be improved by using continuous color schemes for continuous variables (e.g. Figure 1c) and by avoiding the use of rainbow color schemes, which are not color-blindness friendly (Figure 8a and 8b).

*We agree. As suggested by the reviewer, we will replace the color scheme in Figure 1c with a strict continuous one and for Figure 8 a more color-blindness friendly color scheme.*

Furthermore, complete legends should be provided for all figures.

*We are not quite sure what is missing. We agree that some figures of the kind like Figure 2, 4, 6 etc. do not have legends. However, all the relevant information is in the caption, which gives the period, the range and the underlying data for the calculation of the range. This would certainly be too much text for a legend. However, we will add simplifications of all the information in a legend (e.g. red: future climate, blue: current climate).*

**Minor points**

L. l. 40-41: I don't think that it is correct to say that 'no studies have previously included internal climate variability in their analysis' as internal climate variability is per definition part of every change impact assessment. However, it might be ok to say that 'the relative importance of internal variability compared to other uncertainty sources has not been previously assessed.'

*We will follow the suggestions by the reviewer.*

L.58-59: needs rephrasing.

*We will rephrase this sentence.*

L. 61: 'they' will

*Many thanks for finding this error. We will change as suggested.*

L. 63: that 'drive runoff'

*Many thanks for finding this error. We will change as suggested.*

L. 72-73: repeats content already provided in l. 68-69 and can be removed.

*Many thanks for finding this error. We will change as suggested.*

L. 140: How do you define 'frequency', number of events per year or non-exceedance probability or anything else?

*We will add a definition of frequency here, i.e. the number of events per year dependent on an event size larger than a certain threshold, which is equivalent to the yearly exceedance probability.*

L. 259-260: Not sure whether the statement that 'the uncertainty range for current climate is, by definition, only determined by natural variability' is true. If the simulations for the current period are run with different climate models (which is as I understand it the case), climate model uncertainty might also be present.

*We will clarify in the methods section that the current climate period is NOT affected by climate model uncertainty in our model setup.*

L. 279-280: Don't understand this sentence and think it needs rephrasing.

*We will rephrase this sentence.*

L. 293: Sentence seems incomplete. Would be nice to repeat the pixel-based criteria, which the reader might no longer have present at this stage.

*We will do as suggested.*

---

## Author Comment (AC2)

**Reviewer 2**

*We want to thank Reviewer 2 for her/his constructive comments. Our answers are written in italic letters alongside to the reviewer's comments.*

This study assesses projected evolutions of snow-related events in a small alpine region located in Switzerland, using a simulation chain composed of dynamical climate simulations, a stochastic precipitation generator, a snow model. This study provides interesting results about these possible future events, and the methodological choices seem reasonable, at least for the simulations, but there are two main aspects of the manuscript that need to be improved.

*Thank you for your assessment. We agree that the method description needs to be expanded and that the manuscript would benefit from a more chronological sequence with a clearer separation of sections. We will also clearly describe the advantages of the chosen uncertainty partitioning method compared to other methods. For more details, please see our specific responses to your comments below.*

**1. Presentation of the methodology**

Section 2 is difficult to follow for several reasons. The first reason is that the different subsections 2.2, 2.3, 2.4 do not follow a logical order. When the snow model is described, we do not know how its inputs (total precipitation, air temperature, etc.) are obtained, or their spatial resolution. Another example, factors of change are first introduced in Subsection 2.3 whereas they are obtained from climate model outputs in Subsection 2.4. I advise following the order of the simulation chain: 1/Climate models, 2/ Weather generator, 3/ Snow model.

*We will reorder the method section in order following the input of Reviewer 1 and 2*

Secondly, while I understand that all the details of the methodology cannot be provided, the current presentation lacks important information. In particular, from Table 1, it seems that the different precipitation products are used to fit different properties of the precipitation fields (i.e. monthly mean rainfall using optimal interpolated fields, mean areal rainfall using weather radar data). Does it mean that the variability of precipitation at a monthly scale (mean, variance, skewness, etc.) is reproduced using these optimal interpolated fields? What information is used to reproduce statistical properties at a finer resolution (hourly, daily)? For example, how the largest ("extreme") values at daily and sub-daily scales are reproduced? Since this is an important aspect of the study which focuses on intense rain-on-snow events, it needs to be clarified.

*It is correct and useful that multiple sources representing different time scales are used to train the weather generator model. Indeed, the variables listed in Table 1 were used both for calibration and validation of the model (so, yes, for example, the spatial variability of precipitation is reproduced by using the optimal interpolated fields). We agree with the reviewer that additional details need to be provided on the model's ability to reproduce sub-daily climatic variables (such as precipitation, temperature, and radiation) and extremes, and we will include a new figure to illustrate this.*

It was also unclear if there is any information of snow data at a daily scale. To my knowledge, weather radar data do not provide this kind of information. At a monthly scale, it is indicated at l. 110-111 that "optimal interpolation (OI) of snow depth sensor data and a gridded precipitation product, RhiresD) are used, but in Table 1, the line "Optimal interpolated fields" indicates that it is used to fit "Monthly mean rainfall", not snow, so that it is unclear if these OI fields provide total precipitation values or only rainfall. I am not sure where the product RhiresD appears in Table 1. What should be clarified is the list of the statistical properties (statistic, spatial and temporal resolution) of snow and rain that are fitted (and simulated) by the weather generator, and what source of information is used for each of these statistics.

*We did not use daily snow data for training the weather generator. On a monthly level, we did so in the form of updated total precipitation to improve the standard Swiss precipitation product RhiresD, which suffers from a suboptimal station distribution at higher elevations as well as from undercatch of unshielded precipitation gauges. Using the optimal interpolation described in Magnusson et al. (2014), we assimilated daily snow depth measurements using RhiresD as the background field.*

*We will specify the use of snow information in more detail and also include more information about the interpolation of the precipitation used to train the weather generator.*

At l. 131, it is indicated that factors of change are calculated, but no details are provided. For example, the factors of change are usually computed with respect to a reference period, but I could not find this information.

*A control period of 30 years was used to compute the factors of change for mean temperature and precipitation change. We will add this information and explain the factors of change further.*

**2. Uncertainty assessment**

The uncertainty assessment really puzzled me. There is a large number of publications on uncertainty partitioning for climate model simulations (Déqué et al., 2007; Hawkins and Sutton, 2009; Northrop and Chandler, 2014; and many others). These papers all apply an Analysis of Variance (ANOVA) method which provides a clear and rigorous framework in order to obtain a total variance and its components. The different contributions logically sum to one. I do not really understand the approach proposed in Fatichi et al. (2016) which is based on the evaluation of percentile ranges. At l. 154, it is indicated that the 5-95th percentiles obtained from the ten climate models actually refer to the minimum and the maximum, which seems to be a major flaw of the method. Low and high percentiles cannot be obtained from a very limited number of climate simulations (even if you emulate these simulations) and the evaluation of the dispersion (variance) is the best that you can obtain. Secondly, I cannot understand how we can interpret the different contributions if they do not sum to one (l. 167). Fractional uncertainty, as a percentage (e.g. Fig. 3 in Hawkins and Sutton, 2009) provides a direct assessment of the most important contributors to the uncertainty.

*We follow the uncertainty partition method described by Fatichi et al. (2016), which is significantly different from other partition methods (such as the one described by Hawkins and Sutton that the reviewer mentioned). In our case, the reason to use the first method and not the second is that it is more suitable for the situation where you have the complete internal climate variability - model uncertainty - scenario uncertainty chain (as obtained from the AWE-GEN-2d model). We do agree that the method that we used is not described sufficiently (as the first reviewer noted as well) and we will revise the method section to explain the method and how we use quantile ranges. Using Fatichi et al.'s method, the fractional uncertainty contribution cannot be summed to one, but to interpret the different contributions, they do not need to be summed to one. Due to the reviewer comment, we see the need to explain the method and its differences from other uncertainty partitioning methods in more detail. We will therefore extend that section of the manuscript.*

At l. 164-165, it is indicated that "weights [are used] to avoid overweighting days with only low climate change signal uncertainty". I do not see the problem of having a low climate change signal uncertainty, and why it becomes a problem using your approach.

*We have chosen to weight the annual mean of daily fractional uncertainties with daily total uncertainty. This is, in our opinion, meaningful because for days with a low total uncertainty the distribution between the uncertainty sources is of minor importance. These days may, however, substantially influence this distribution in a non-weighted annual mean. We will clarify this procedure in more detail in the revised manuscript.*

For all these reasons, I strongly recommend using a standard ANOVA approach for the uncertainty assessment.

**3. Minor comments:**

All figure captions: Usually "(a)", "(b)", etc. are placed before the description of the

respective subpanels.

*We will change this as suggested.*

Figure 2: The labels of the y-axis are ILWR and ISWR for panels (c) and (d) whereas in

the caption, it is inverted.

*Many thanks for finding this error. We will change this as suggested.*

260: "by definition, only determined by natural variability": I am not sure what you

mean by "definition". There is also an important part of model uncertainty for the

current climate periods. This kind of uncertainty is usually removed mechanically using

factors of change (as you did probably). A clarification would be appreciated here.

*We will clarify in the revised manuscript that current climate periods are only affected by natural climate variability in our model setup.*

14 – l. 293: I guess "Figure 7" is missing.

*Many thanks for finding this error. We will change this as suggested.*

17: Figure 9 is not presented and described.

*Many thanks for finding this error. We will change this as suggested.*

**References**

Déqué, M., D. P. Rowell, D. Lüthi, F. Giorgi, J. H. Christensen, B. Rockel, D. Jacob, E.

Kjellström, M. de Castro, and B. van den Hurk. 2007. "An Intercomparison of Regional

Climate Simulations for Europe: Assessing Uncertainties in Model Projections." *Climatic*

*Change* 81 (1): 53–70. https://doi.org/10.1007/s10584-006-9228-x.

Hawkins, E., and R. Sutton. 2009. "The Potential to Narrow Uncertainty in Regional

Climate Predictions." *Bulletin of the American Meteorological Society* 90 (8): 1095–1107.

https://doi.org/10.1175/2009BAMS2607.1.

---

## Author Response (AR1)

Dear Editor and dear Reviewers

The following document contains the point-by-point response to the reviews.

The most important changes are:

1) Reorganization of the manuscript structure

2) Changed the partitioning method from Fatichi et al. 2016 to Yip et al. 2011, which is an ANOVA method with interaction term

3) Added details to the methods section, in particular the weather generator and the data used for calibration

4) Added a section on limitations and generalization

5) Revised the figures

Best regards

Michael Schirmer

**Answer to Reviewer 1**

*We want to thank Reviewer 1 for her/his constructive comments. Our answers are written in italic letters alongside to the reviewer's comments.*

**Summary**

The authors use a model chain consisting of climate models, a weather generator, and an energy balance snow model to identify dominant uncertainty sources in future changes in snow-water-equivalent and rain-on-snow runoff. They show that changes in ROS events emerge till the end of the century despite large uncertainties while ROS events with substantial snowmelt contributions don't show a clear change signal.

**General remarks**

The study by Schirmer et al. builds on a complex model chain consisting of climate models, a weather generator and an energy balance snow model to assess the importance of internal variability on the detection of future changes in snow and rain-on-snow runoff events. I think that the combination of different model types to better describe internal variability is a generally a valid approach to determine the importance of internal variability in change assessments of snow-related quantities compared to other uncertainty sources. However, I see a substantial need for clarification regarding the research questions and methodology and think that the approach chosen to decompose uncertainty into different contributors needs refinement. Given the current 'incomplete' methods descriptions, it is difficult to assess the validity of the results. Furthermore, I think that the manuscript would profit from reorganization, i.e. restructuring the methods section following a more logical sequence and from separating the results from the discussion. Finally, the manuscript would in my opinion profit from a visualization of the most important modeling steps and their relationships

and from refining figures by adapting color schemes and adding legends. Please find my more detailed review below.

*Thank you for your assessment. We have agreed on these main topics and added more details on the method description and a flow chart. We have separated the sections more clearly and put them in chronological order. We have also refined the illustrations. We also decided to change the partitioning method. For more details, see our specific responses to your comments below.*

**Major points**

1. Research questions: the research questions are not entirely clear and should be explicitly stated in the introduction. From how I understand the study it is something along the lines of: 'How does the importance of internal variability differ between temperature-driven snow resources and rain-driven rain-on-snow events' and 'When is the time of emergence of changes in snow availability and rain-on-snow runoff.'

*We reworked the final paragraph of the introduction to ensure that the research questions are concisely formulated.*

2. Introduction: In addition to model-based studies looking at changes in rain-on-snow floods, there are also observation-based studies, which I think should be mentioned in the introduction. E.g. Sikorska and Seibert (2020; 10.1080/02626667.2020.1749761) or Cheggwidden et al. (2020; 10.1088/1748-9326/ab986f).

*We agree and added these studies in the introduction.*

3. Methods section organization: The methods section does not seem to follow a logical order and could in my opinion be more logically organized by following a 'chronological' modeling order. E.g. Area, Climate models, Weather generator, Snow model, Rain-on-snow definition, Change assessment, Uncertainty decomposition.

*Our original methods section was structured so that the most important methods for this manuscript are described first (i.e. the weather generator and the snow model). However, we put the methods in a more chronological order, as suggested, to clarify the methodological flow.*

Providing a flowchart linking the most important modeling and analysis steps might enable further improvements in communication. Furthermore, the methods section lacks important methodological detail, which makes it difficult to assess the validity of the results.

*We agree and added a flowchart as well as more details in the methods sections.*

4. Glacier retreat: The study region is influenced by a glacier, which affects runoff formation. However, the glacier-related changes in flow are not represented in the modeling chain (l. 86-87). This does not seem to be justified and might explain why melt-influenced changes in ROS events are don't show up clearly.

*Yes, this study region is influenced by a large glacier. However, we do not want to investigate the combined effects of snow and glacier retreat on mean snow water resources or ROS properties, but only to show the effects of snow. For this reason, we have not compared observed runoff measurements in the area with our modelled data. We chose this catchment because of its good data availability and high elevation. The study area is treated as if there were no glacier. Only the altitudes and the meteorological input data play a role for the study area. It can therefore be considered rather as an example area, perhaps even a virtual area, for which the influence of natural climate*

*variability on snow water resources and ROS events is studied as an example. These considerations are now presented in more detail.*

5. Weather generator: The weather generator description (Section 2.3) lacks important detail and it is therefore difficult to assess the validity of the approach. E.g. how does the weather generator use the climate simulations, how does the weather generator work, how is the temporal downscaling performed (l. 122), how are the different variables generated (l. 122), how is the inter-variable consistence (l.124) evaluated?

*In the new version we have ensured that the references for these important questions are clearly visible, but also included more details from cited publications. There are two comprehensive publications, Peleg et al. (2017, 2019), which we cited in first version of manuscript to answer these questions. We understand that a manuscript should stand on its own, but we also had to consider the readability of this already quite long manuscript. This is especially true since this study is an application to the weather generator among other methods. We have also not gone into detail on energy balance snow modelling or climate modelling. Given the complexity of, for example, how the weather generator works in detail, we have chosen in the original manuscript to provide references to these issues, but at the same time to show in broad terms how the weather generator was trained and applied.*

6. Snow model and variables: It remains unclear to me how the weather generator output is used to derive different snow-related variables (Section 2.2). Was the analysis performed per grid cell? Which variables were exactly derived? How was the model calibrated (l.95)? And what does the 'unpublished' model adjustment (l. 95-96) do?

*We apologize and agree that these points should be explained more clearly. In the more chronological order of the methods section suggested above we have included more details answering these questions. The results achieved with the snow model are mean values either within the whole catchment or within all ROS affected pixel. This is described in each results section and is placed also in the method section.*

*The model is an open-source energy balance snow model, which is publicly available. These types of models are usually not calibrated, and we did not calibrate our model as well. We provided more details on the adjustments to the model.*

7. Bias correction: Section 2.5. suggests that some bias correction might have been necessary to adjust simulated to observed values. Was such bias correction performed and if so why?

*Yes, the climate model chains were bias corrected and downscaled with quantile mapping within the official Swiss climate scenarios CH2018 (CH2018 Project Team, 2018). However, for the factor of change approach to re-parameterize the weather generator (which was trained beforehand to simulate the current climate conditions) the bias correction has only a minor impact compared to directly applying debiased climate model output. A climate change signal can be changed due to quantile mapping compared to raw climate model outputs, however, it was shown that these changes were small (CH2018, Technical Report)*

8. Uncertainty partitioning: The uncertainty partitioning procedure described in Section 2.6. does not seem to properly separate internal variability (residuals) from the signal. Or at least I cannot see how the different uncertainty components have been decomposed e.g. using a procedure such as the one proposed by Hawkins and Sutton (2009; 10.1175/2009BAMS2607.1). The procedure used to derive climate model uncertainty also seems to encompass internal variability (l.153-154) and the procedure used to derive internal variability also seems to include climate model uncertainty (l.156-

157). Furthermore, it would be nice to compute fractional uncertainty contributions that add up to 1, which currently does not seem to be the case.

*After applying and comparing the methods of Lehner et al. (2020) and Yip et al. (2011) and discussing the comments of the two reviewers, we decided to use the ANOVA method of Yip et al. (2011). This method is applicable to data where the complete chain of internal climate variability - model uncertainty - scenario uncertainty is available, it is able to quantify the covariance between the latter two sources of uncertainty, and it is able to provide fractional uncertainties that sum to one. The ability to combine the latter two features was an important argument for us to change the partitioning method, as we are aware of the potential for misinterpretation if the fractional uncertainties do not sum to one. The method of Yip et al. (2011) is also more widely used and more directly comparable to the original work of Hawkins and Sutton (2009), so our results can be interpreted in a broader context.*

*The dominance of natural climate variability was confirmed with the new method, with one exception, namely for delta-SWE at the end of the century. This is the only example where scenario uncertainty was a significant contributor even in the original method used. In contrast to the comments of reviewer 2, who saw the biggest problem in using only 10 climate models, we see - in this example - the biggest differences in scenario uncertainty with only two values. It makes a big difference whether one estimates the relative dispersion of two values versus those of 10 and 50 values with the variance (even without scaling the variance with N-1, but with N as in Yip et al. (2011)) or with the quantile range. Due to the quadratic behavior of the variance, this difference is not significant in the other examples where scenario uncertainty does not play a major role in either method. We generally see lower percentages for the contribution of natural climate variability with the new method.*

*Further differences result from our new decision to show only monthly averages instead of daily distribution of uncertainties (new Figure 11). We think that a decision maker is not interested in how the uncertainty is divided for daily SWE values. Natural climate variability is smaller relative to the other sources for monthly mean values than for daily values. However, the most important change resulted from the switch from quantile ranges to variances.*

9. Validation: I think that the methods section needs a 'Validation' subsection describing how the different models were evaluated. E.g. how were the validation stations chosen? Which variables were validated, …

*We added a verification subsection in the methods section in order to be clearer on this topic.*

10. Rain-on-snow events: how have these events been defined? There is a section called 'rain-on-snow' definition, which does, however, not really explain what you understand by a 'rain-on-snow' event. How is the 'surface water input' computed?

*We do not define ROS events, but apply daily pixel-based criteria that then result in a certain size of "contributing area" each day. These criteria are defined in Table 2. An "ROS day" can then be defined as a day with a contributing area of a certain size, which may depend on the application or the user. We determined a climate change signal of ROS frequency with variable event sizes (x-axis of Figure 7, now 8). The intensity and contribution of snowmelt required a certain size of event (> 1/3 of the total area affected according to pixel-based criteria). Since the reviewers have pointed out that methods and results should be more clearly separated, we moved this information from the results to the method section.*

*The "surface water input" is calculated with the energy balance snow model and is the water input available at the ground surface through either snowpack runoff, or rain in case of snow-free conditions, or a mixture of both in case of fractional snowcover. We added this definition in the methods section.*

11. Results: I would clearly separate the results part from the methods section and discussion. Some parts can be moved from the Results to the Methods section (e.g. l. 206-216) and other parts to a newly created Discussion section (essentially everything that compares the study's findings to findings of existing studies). Furthermore, it would be nice if the results section followed a similar structure as the methods section.

*As already written above, we added more details in the methods section, which will also help to separate results from methods. We thank the reviewer for this suggestion. However, we want to keep the results and discussion part combined ["… because readers can seldom make sense of results alone without accompanying interpretation — they need to be told what the results mean"]. https://www.nature.com/scitable/topicpage/scientific-papers-13815490/*

12. Figure 11: Would be nice to depict the fractional contributions of the individual uncertainty sources. Currently, all of the sources seem to not be clearly separated (probably also related to the issue risen in comment 8).

*Due to the new partitioning method, the new Figure 11 is now showing both the absolute and fractional contribution of each uncertainty source. Because of the new method the sources are now clearly separated.*

13. Discussion: I miss a discussion of alternative strategies that could be used to quantify the relative importance of internal variability to other uncertainty sources besides weather generators. Recently, quite a few studies have used single model initialized large ensembles (SMILES) for uncertainty decomposition and I think it would be important to mention this (e.g. Maher et al. 2021; 10.5194/esd-12-401-2021 or Lehner et al. 2020; 10.5194/esd-11-491-2020). Other topics I would address in the discussion section include: model deficiencies, comparisons of results with other studies (as distributed throughout the results section), and the generalizability of the findings to other geographical and climatic contexts given that the study relies on one catchment only.

*We added a discussion on other ways to quantify natural climate variability, along with the suggested publications. The findings are also discussed with regard to model deficiencies and generalizability.*

14. Figures: Figure design should be improved by using continuous color schemes for continuous variables (e.g. Figure 1c) and by avoiding the use of rainbow color schemes, which are not color-blindness friendly (Figure 8a and 8b).

*We apologize and agree. As suggested by the reviewer, we replaced the color scheme in Figure 1c with a strict continuous one and for the new Figure 9 a more color-blindness friendly color scheme.*

Furthermore, complete legends should be provided for all figures.

*We added legends to all figures.*

**Minor points**

L. l. 40-41: I don't think that it is correct to say that 'no studies have previously included internal climate variability in their analysis' as internal climate variability is per definition part of every change impact assessment. However, it might be ok to say that 'the relative importance of internal variability compared to other uncertainty sources has not been previously assessed.'

*We followed the suggestions by the reviewer.*

L.58-59: needs rephrasing.

*We chose to delete these details when citing this study.*

L. 61: 'they' will

*We changed as suggested.*

L. 63: that 'drive runoff'

*We changed as suggested.*

L. 72-73: repeats content already provided in l. 68-69 and can be removed.

*One is about SWE in comparison to runoff and one is ROS frequency in comparison to SWE. We reformulated this section.*

L. 140: How do you define 'frequency', number of events per year or non-exceedance probability or anything else?

*We added a definition of frequency here, i.e. the number of events per year dependent on an event size larger than a certain threshold, which is equivalent to the yearly exceedance probability.*

L. 259-260: Not sure whether the statement that 'the uncertainty range for current climate is, by definition, only determined by natural variability' is true. If the simulations for the current period are run with different climate models (which is as I understand it the case), climate model uncertainty might also be present.

*We clarified in the methods section that the current climate period is NOT affected by climate model uncertainty in our model setup. The added flow chart also supports this understanding.*

L. 279-280: Don't understand this sentence and think it needs rephrasing.

*We rephrased this sentence.*

L. 293: Sentence seems incomplete. Would be nice to repeat the pixel-based criteria, which the reader might no longer have present at this stage.

*However, we believe that it is not good for the flow of reading to repeat the criterion here and preferred a reference to the relevant table.*

**Reference**

CH2018 (2018) CH2018 – Climate Scenarios for Switzerland, Technical Report, National Centre for Climate Services, Zurich, 271 pp.

**Answer to Reviewer 2**

*We want to thank Reviewer 2 for her/his constructive comments. Our answers are written in italic letters alongside to the reviewer's comments.*

This study assesses projected evolutions of snow-related events in a small alpine region located in Switzerland, using a simulation chain composed of dynamical climate simulations, a stochastic precipitation generator, a snow model. This study provides interesting results about these possible future events, and the methodological choices seem reasonable, at least for the simulations, but there are two main aspects of the manuscript that need to be improved.

*Thank you for your assessment. We agree that the method description had to be expanded and that the manuscript benefits from a more chronological sequence with a clearer separation of sections. We also decided to change the partitioning method. For more details, please see our specific responses to your comments below.*

**1. Presentation of the methodology**

Section 2 is difficult to follow for several reasons. The first reason is that the different subsections 2.2, 2.3, 2.4 do not follow a logical order. When the snow model is described, we do not know how its inputs (total precipitation, air temperature, etc.) are obtained, or their spatial resolution. Another example, factors of change are first introduced in Subsection 2.3 whereas they are obtained from climate model outputs in Subsection 2.4. I advise following the order of the simulation chain: 1/Climate models, 2/ Weather generator, 3/ Snow model.

*We reordered the method section in order following the input of Reviewer 1 and 2*

Secondly, while I understand that all the details of the methodology cannot be provided, the current presentation lacks important information. In particular, from Table 1, it seems that the different precipitation products are used to fit different properties of the precipitation fields (i.e. monthly mean rainfall using optimal interpolated fields, mean areal rainfall using weather radar data). Does it mean that the variability of precipitation at a monthly scale (mean, variance, skewness, etc.) is reproduced using these optimal interpolated fields? What information is used to reproduce statistical properties at a finer resolution (hourly, daily)? For example, how the largest ("extreme") values at daily and sub-daily scales are reproduced? Since this is an important aspect of the study which focuses on intense rain-on-snow events, it needs to be clarified. […] What should be clarified is the list of the statistical properties (statistic, spatial and temporal resolution) of snow and rain that are fitted (and simulated) by the weather generator, and what source of information is used for each of these statistics.

*It is correct and useful that multiple sources representing different time scales are used to train the weather generator model. The model is calibrated at sub-daily scales as explained below; monthly statistics are only used to demonstrate the model abilities to reproduce precipitation patterns properly. The procedure of the precipitation calibration and modeling is detailed explained in Peleg et al. (2017), hence we added only a concise summary in the manuscript. In brief: the spatial structure of the precipitation fields and its areal statistics (i.e. the average precipitation over the entire domain and the fraction of area affected by precipitation) are derived from the weather radar at intervals of 5 min. To determine the lengths of the storms and the dry periods we used information not from the radar data (7 years) but from a nearby station (Grimsel, almost 30 years). As the precipitation intensities from the weather radar are associated with high uncertainties, we apply then a correction to the precipitation intensities for each grid cell at the domain using gridded precipitation product from MeteoSwiss (RhiresD). To demonstrate the ability of the model to reproduce extreme*

*precipitation (on a daily time scale) we added a new figure in the supplementary material (Figure S2). We also added the spatial and temporal resolutions in Table 1 for all data sources.*

It was also unclear if there is any information of snow data at a daily scale. To my knowledge, weather radar data do not provide this kind of information. At a monthly scale, it is indicated at l. 110-111 that "optimal interpolation (OI) of snow depth sensor data and a gridded precipitation product, RhiresD) are used, but in Table 1, the line "Optimal interpolated fields" indicates that it is used to fit "Monthly mean rainfall", not snow, so that it is unclear if these OI fields provide total precipitation values or only rainfall. I am not sure where the product RhiresD appears in Table 1.

*We did not use daily snow data for training the weather generator, and also not achieved from weather radar. On an annual level (not on a monthly level as wrongly indicated in the first version of the manuscripot), we did used daily station data in the form of updated total precipitation (we also changed the naming from rainfall to precipitation in Table 1) to improve the standard Swiss precipitation product RhiresD, which suffers from a suboptimal station distribution at higher elevations as well as from undercatch of unshielded precipitation gauges. Using the optimal interpolation described in Magnusson et al. (2014), we assimilated daily snow depth measurements using RhiresD as the background field.*

At l. 131, it is indicated that factors of change are calculated, but no details are provided. For example, the factors of change are usually computed with respect to a reference period, but I could not find this information.

*A control period of 30 years was used to compute the factors of change for mean temperature and precipitation change. We added this information and explain the factors of change further.*

**2. Uncertainty assessment**

The uncertainty assessment really puzzled me. There is a large number of publications on uncertainty partitioning for climate model simulations (Déqué et al., 2007; Hawkins and Sutton, 2009; Northrop and Chandler, 2014; and many others). These papers all apply an Analysis of Variance (ANOVA) method which provides a clear and rigorous framework in order to obtain a total variance and its components. The different contributions logically sum to one. I do not really understand the approach proposed in Fatichi et al. (2016) which is based on the evaluation of percentile ranges. At l. 154, it is indicated that the 5-95th percentiles obtained from the ten climate models actually refer to the minimum and the maximum, which seems to be a major flaw of the method. Low and high percentiles cannot be obtained from a very limited number of climate simulations (even if you emulate these simulations) and the evaluation of the dispersion (variance) is the best that you can obtain. Secondly, I cannot understand how we can interpret the different contributions if they do not sum to one (l. 167). Fractional uncertainty, as a percentage (e.g. Fig. 3 in Hawkins and Sutton, 2009) provides a direct assessment of the most important contributors to the uncertainty.

*After applying and comparing the methods of Lehner et al. (2020) and Yip et al. (2011) and discussing the comments of the two reviewers, we decided to use the ANOVA method of Yip et al. (2011). This method is applicable to data where the complete chain of internal climate variability - model uncertainty - scenario uncertainty is available, it is able to quantify the covariance between the latter two sources of uncertainty, and it is able to provide fractional uncertainties that sum to one. The ability to combine the latter two features was an important argument for us to change the partitioning method, as we are aware of the potential for misinterpretation if the fractional uncertainties do not sum to one. The method of Yip et al. (2011) is also more widely used and more directly comparable to the original work of Hawkins and Sutton (2009), so our results can be interpreted in a broader context.*

*The dominance of natural climate variability was confirmed with the new method, with one exception, namely for delta-SWE at the end of the century. This is the only example where scenario uncertainty was a significant contributor even in the original method used. We see - in this example - the largest differences in scenario uncertainty with only two values. It makes a big difference whether one estimates the relative dispersion of two values versus those of 10 and 50 values with the variance (even without scaling the variance with N-1, but with N as in Yip et al. (2011)) or with the quantile range. Due to the quadratic behavior of the variance, this difference is not significant in the other examples where scenario uncertainty does not play a major role in either method. We generally see lower percentages for the contribution of natural climate variability with the new method.*

*Further differences result from our new decision to show only monthly averages instead of daily distribution of uncertainties (new Figure 11). We think that a decision maker is not interested in how the uncertainty is divided for daily SWE values. Natural climate variability is less large relative to the other sources for monthly mean values than for daily values. However, the most important change resulted from the switch from quantile ranges to variances.*

At l. 164-165, it is indicated that "weights [are used] to avoid overweighting days with only low climate change signal uncertainty". I do not see the problem of having a low climate change signal uncertainty, and why it becomes a problem using your approach.

*The use of weights was no longer needed, as in the new Figure 11 we now show monthly data of partial uncertainties, which we no longer average to an annual value.*

For all these reasons, I strongly recommend using a standard ANOVA approach for the uncertainty assessment.

*As described above, we now follow this suggestion.*

**3. Minor comments:**

All figure captions: Usually "(a)", "(b)", etc. are placed before the description of the

respective subpanels.

*We changed this as suggested.*

Figure 2: The labels of the y-axis are ILWR and ISWR for panels (c) and (d) whereas in

the caption, it is inverted.

*Many thanks for finding this error. We changed this as suggested.*

260: "by definition, only determined by natural variability": I am not sure what you

mean by "definition". There is also an important part of model uncertainty for the

current climate periods. This kind of uncertainty is usually removed mechanically using

factors of change (as you did probably). A clarification would be appreciated here.

*We clarified in the revised manuscript that current climate periods are only affected by natural climate variability in our model setup. The newly added flow chart should also contribute to this understanding.*

14 – l. 293: I guess "Figure 7" is missing.

*Many thanks for finding this error. We changed this as suggested.*

17: Figure 9 is not presented and described.

*Many thanks for finding this error. We moved this figure to the Supplement in order to reduce the number of figures in the main document.*

**References**

Déqué, M., D. P. Rowell, D. Lüthi, F. Giorgi, J. H. Christensen, B. Rockel, D. Jacob, E.

Kjellström, M. de Castro, and B. van den Hurk. 2007. "An Intercomparison of Regional

Climate Simulations for Europe: Assessing Uncertainties in Model Projections." *Climatic

Change* 81 (1): 53–70. https://doi.org/10.1007/s10584-006-9228-x.

Hawkins, E., and R. Sutton. 2009. "The Potential to Narrow Uncertainty in Regional

Climate Predictions." *Bulletin of the American Meteorological Society* 90 (8): 1095–1107.

https://doi.org/10.1175/2009BAMS2607.1.

---

## Author Response (AR2)

*We want to thank the reviewer for her/his constructive comments. Our answers are written in italic letters alongside to the reviewer's comments.*

I thank the authors for the modifications made to the paper, which follow the different recommendations made by the reviewers. They have greatly improved the presentation and the understanding of the results. I have a few remarks that should probably be addressed before publication.

Natural variability

Throughout the paper, the natural variability is assumed to be adequately and fairly represented by the weather generator. However, as I understand it, the weather generator only represents the variability related to the purely stochastic behavior of the meteorological process and describes the high-frequency component of the internal variability. The low-frequency variability describes in Deser et al. (2012b; Climate Dynamics) as the "variability [that] arises from processes internal to the coupled ocean-atmosphere system via dynamic and thermodynamic interactions" is not reproduced by the weather generator. This point should be mentioned and discussed in Section 3.3.

*We thank the reviewer for this comment and included this point in Section 3.3 (565ff).*

Factors of change

There is not much explanation about the factors of change, but they seem to be a very important component of this study. As I understand it, factors of change are mean temperature and precipitation change are absolute differences of annual mean temperature obtained from the RCM runs, between the historical and future periods. I assume that relative differences are computed for precipitation changes. An equation in Section 2.2.1 would clarify these points.

*We added equations Eq (1) and (2) in the appendix and included more details of the use of factors of change in section 2.2.1 (line 107ff).*

I also imagine that these absolute and relative differences are applied to some parameters of the stochastic generator. At l. 144, the authors indicate "a reparametrized setup using factors of change (FC)" but we do not know if the factor of changes only affects the annual mean temperatures and precipitations in the simulations from the weather generator or other aspects. Additional details are thus required to understand how these factors of change are applied since it is the only part of the simulation chain that actually leads to different climate change responses. A figure showing the factors of change that have been obtained would certainly help to understand their impact.

*We added information which variables were directly changed by the FC approach and mention the interdependency between variables in the weather generator, which indirectly affects other variables (line 147ff). We aimed to provide a figure in the Supplement similar as in Peleg et al. (2019), Fig. 3b, which shows the factors of change for a nearby study area. However, the time restrictions did not allow us to rerun the code as this information is not just an external data set. Thus, we chose to reference Figure 3b in Peleg et al. (2019). If the Editor feels that this additional figure in the Supplement is crucial for publication, we will provide this information with a bit of extra time.*

Correction factor

l. 186: "a correction factor of 1.3" -> This precision should not appear in the section "verification" as this affects the input. How the factor of 1.3 is chosen? Is it applied to all grid cells? Is there a publication to

justify this choice of 1.3? There are often gradients that are applied for this correction as a function of the elevation, but a correction of +30% is very large.

*For unshielded precipitation gages as those operated by MeteoSwiss in Switzerland snowfall site specific undercatch corrections of 30% are (unfortunately) quite normal, see e.g. Egli et al. (2009). Although section 2.3 is entitled "verification" we note the correction factor as part of the description for the input data of the snow model simulations presented in Figure 4. Figure 4 is intended to demonstrate the model's general ability to represent the temporal snow accumulation and melt dynamics, where using local input data (affected by undercatch) necessitated the application of a local undercatch correction. This, of course, was not the case for all the other snow model simulations where the weather generator provided model input, which we now explicitly state in the text (line 192ff)*

Minor comments

l. 153: "provide hourly data" -> could you be more specific? For example, "areal precipitation and temperature data at an hourly scale".

*We combined item (1) and (2) into on item to clarify that the weather generator is used for providing hourly data for the full set of required inputs for the energy balance snow model, which are described in the next section (line 158ff).*

l. 212: "The intensity and contribution of snowmelt required to define a size of a ROS event" -> A part of the sentence seem to be missing.

*This sentence was corrected (line 219f).*

*References (now also part of the manuscript)*

*Egli, L., Jonas, T., Meister, R.: Comparison of different automatic methods for estimating snow water equivalent. Cold Regions Science and Technology, 57, 107-115, https://doi.org/10.1016/j.coldregions.2009.02.008, 2009.*